# Addressing Bias in Online Selection with Limited Budget of Comparisons

**Ziyad Benomar**
ENSAE, Ecole Polytechnique,
FairPlay joint team
ziyad.benomar@ensae.fr

**Evgenii Chzhen**
CNRS, LMO, Université Paris-Saclay
evgenii.chzhen@universite-paris-saclay.fr

**Nicolas Schreuder**
CNRS, Laboratoire d'informatique
Gaspard Monge (LIGM/UMR 8049)
nicolas.schreuder@cnrs.fr

**Vianney Perchet**
CREST, ENSAE, Criteo AI LAB
Fairplay joint team
vianney.perchet@normalesup.org

## Abstract

Consider a hiring process with candidates coming from different universities. It is easy to order candidates with the same background, yet it can be challenging to compare them otherwise. The latter case requires additional costly assessments, leading to a potentially high total cost for the hiring organization. Given an assigned budget, what would be an optimal strategy to select the most qualified candidate? We model the above problem as a multicolor secretary problem, allowing comparisons between candidates from distinct groups at a fixed cost. Our study explores how the allocated budget enhances the success probability in such settings.

## 1   Introduction

Online selection is among the most fundamental problems in decision-making under uncertainty, Multiple problems within this framework can be modeled as variants of the secretary problem [Dynkin, 1963, Chow et al., 1971], where the decision-maker has to identify the best candidate among a pool of totally ordered candidates, observed sequentially in a uniformly random order. When a new candidate is observed, the decision maker can either select them and halt the process or reject them irrevocably. The optimal strategy is well known and consists of skipping the first $1/e$ fraction of the candidates and then selecting the first candidate that is better than all previously observed ones. This strategy yields a probability $1/e$ of selecting the best candidate. A large body of literature is dedicated to the secretary problem and its variants, we refer the interested reader to [Chow et al., 1971, Lindley, 1961] for a historical overview of this theoretical problem.

In practice, as pointed out by several social studies, the selection processes often do not reflect the actual relative ranks of the candidates and might be biased with respect to some socioeconomic attributes [Salem et al., 2022, Raghavan et al., 2020]. To tackle this issue, several works have explored variants of the secretary problem with noisy or biased observations [Salem and Gupta, 2019, Freij and Wästlund, 2010]. In particular, Correa et al. [2021a] studied the *multi-color secretary problem*, where each candidate belongs to one of $K$ distinct groups, and only candidates of the same group can be compared. This corresponds for example to the case of graduate candidates from different universities, where the within-group orders are freely observable and can be trusted using a metric such as GPA, but inter-group order cannot be obtained by the same metric. This model, however, is too pessimistic, as it overlooks the possibility of obtaining inter-group orders at some cost, through testing and examination. Taking this into account, we study the multicolor secretary problem with a budget for comparisons, where comparing candidates from the same group is free, and comparing candidates

38th Conference on Neural Information Processing Systems (NeurIPS 2024).

from different groups has a fixed cost of 1. We assume that the decision-maker is allowed at most $B$ comparisons. This budget $B$ represents the amount of time/money that the hiring organization is willing to invest to understand the candidate's "true" performance. As in the classical secretary problem, an algorithm is said to have *succeeded* if the selected candidate is the best overall, otherwise, it has *failed*. The objective is to design algorithms that maximize the probability of success.

## 1.1 Contributions

The paper studies an extension of the *multi-color* secretary problem [Correa et al., 2021a], where comparing candidates from different groups is possible at a cost. This makes the setting more realistic and paves the way for more practical applications, but also introduces new analytical challenges.

In Section 2, we describe a general class of Dynamic-Threshold (DT) algorithms, defined by distinct acceptance thresholds for each group, that can change over time depending on the available budget. First, we examine a particular case where all thresholds are equal, which can be viewed as an extension of the classical $1/e$-strategy. However, the analysis is intricate due to additional factors such as group memberships, comparison history, and the available budget. By carefully controlling these parameters, we compute the asymptotic success probability of the algorithm, demonstrating its extremely rapid convergence to the upper bound of $1/e$ when the budget increases, hence constituting a first efficient solution to the problem.

Subsequently, our focus shifts to the case of two groups, where we explore another particular case of DT algorithms: static double threshold algorithms. These involve different acceptance thresholds for each group, that depend on the groups' proportions and the initial budget, but do not vary during the execution of the algorithm. We prove a recursive formula for computing the resulting success probability, and we exploit it to establish a closed-form lower bound and compute explicit thresholds.

In the two-group scenario, we also derive the optimal algorithm among those that do not utilize the history of comparisons, which we refer to as *memory-less*. We present an efficient implementation of this algorithm and demonstrate, via numerical simulations, that it belongs to the class of DT algorithms when the number of candidates is large. Leveraging this insight, we numerically compute optimal thresholds for the two-group case.

## 1.2 Related work

**The secretary problem** The secretary problem was introduced by Dynkin [1963], who proposed the $1/e$-threshold algorithm, having a success probability of $1/e$, which is the best possible. Since then, the problem has undergone extensive study and found numerous applications, including in finance [Hlynka and Sheahan, 1988], mechanism design [Kleinberg, 2005, Hajiaghayi et al., 2007], Nested Rollout Policy Adaptation (NRPA) [Dang et al., 2023], active learning [Fujii and Kashima, 2016], and the design of interactive algorithms [Sabato and Hess, 2016, 2018]. Moreover, the secretary problem has multiple variants [Karlin and Lei, 2015, Bei and Zhang, 2022, Assadi et al., 2019, Keller and Geißer, 2015], and has inspired other works, for instance, related to matching [Goyal, 2022, Dickerson et al., 2019] or ranking [Jiang et al., 2021]. A closely related problem is the prophet inequality [Krengel and Sucheston, 1977, Samuel-Cahn, 1984], where the decision-maker sequentially observes values sampled from known distributions, and its reward is the value of the selected item, in opposite the secretary problem where the reward is binary: 1 if the selected value is the maximum and 0 otherwise. Prophet inequalities also have many applications [Kleinberg and Weinberg, 2012, Chawla et al., 2010, Feldman et al., 2014] and have been explored in multiple variants [Kennedy, 1987, Azar et al., 2018, Bubna and Chiplunkar, 2023, Benomar et al., 2024].

**Different information settings** In some practical scenarios, the secretary problem may present a pessimistic model. Therefore, variants with additional information have been studied. For example, Gilbert and Mosteller [2006] explored a scenario where candidates' values are independently drawn from a known distribution. Other studies have examined potential improvements with other types of information, such as samples [Correa et al., 2021b] or machine-learned advice [Antoniadis et al., 2020, Dütting et al., 2021, Benomar and Perchet, 2023]. Conversely, some works more closely aligned with ours have investigated more constrained settings. Notably, Correa et al. [2021a] introduced the multi-color secretary problem, where totally ordered candidates belong to different groups, and only the partial order within each group, consistent with the total order, can be accessed. Under fairness

constraints, they designed an asymptotically optimal strategy for selecting the best candidate. Other settings with only partial information have been studied as well. For example, Monahan [1980, 1982] addressed the optimal stopping of a target process when only a related process is observed, and they designed mechanisms for acquiring information from the target process. However, these works do not assume a fixed budget and instead consider a penalized version of the problem.

**Online algorithms with limited advice** This paper also relates to other works on online algorithms, where the decision-maker is allowed to query a limited number of hints during execution. The objective of these analyses is to measure how the performance improves with the number of permitted hints. Such settings have been studied, for example, in online linear optimization [Bhaskara et al., 2021], caching, [Im et al., 2022], paging [Antoniadis et al., 2023], scheduling [Benomar and Perchet], metrical task systems [Sadek and Elias, 2024], clustering [Silwal et al., 2023], and sorting [Bai and Coester, 2024, Benomar and Coester, 2024]. Another related paper by Drygala et al. [2023] studies a penalized version of the Bahncard problem with costly hints.

## 2 Formal problem

We consider a strictly totally ordered set of cardinal $N$, whose elements will be called *candidates*. We assume that these candidates are observed in a uniformly random arrival order $(x_1, \ldots x_N)$, and that they are partitioned into $K$ groups $G^1, \ldots, G^k$. For all $t \in [N]$, we denote by $g_t \in [K]$ the group of $x_t$, i.e. $x_t \in G^{g_t}$, and we assume that $\{g_t\}_{t \in [N]}$ are mutually independent random variables

$$\mathbf{P}(g_t = k) = \lambda_k, \quad \forall t \in [N], \forall k \in [K] \ ,$$

where $\lambda_k > 0$ for all $k \in [K]$ and $\sum_{k=1}^{K} \lambda_k = 1$.

We assume that comparing candidates of the same group is free, while comparing two candidates of different groups is costly. To address the latter case, we consider that a budget $B \geq 0$ is given for comparisons and we propose two models: the algorithm can pay a cost of 1 in order to:

1. compare a two already observed candidates $x_t$ and $x_s$ belonging to different group,

2. determine if the current candidate is the best candidate seen so far among all the groups.

For simplicity, we focus on the second model. However, we explain during the paper how our algorithms adapt to the first model and the cost they incur.

When a new candidate arrives, the algorithm can choose to select them, halting the process, or it can choose to skip them, moving on to the next one—hoping to find a better candidate in the future. Once a candidate has been rejected, they cannot be recalled—the decisions are irreversible. Given the total number of candidates $N$, the probabilities $(\lambda_k)_{k \in [K]}$ characterizing the group membership, and a budget $B$, the goal is to derive an algorithm that maximizes the probability of selecting the best overall candidate. We refer to the problem as the $(K, B)$-secretary problem

### 2.1 Additional notation

For all $t < s \in [N]$, we denote by $x_{t:s} := \{x_t, \ldots, x_s\}$, and for all $k \in [K]$ we denote by $G_{t:s}^k$ the set of candidates of group $G^k$ observed between steps $t$ and $s$,

$$G_{t:s}^k := \{x_r \ : \ t \leq r \leq s \text{ and } g_r = k\} = x_{s:t} \cap G^k \ .$$

If $t = 1$, then we lighten the notation $G_s^k := G_{1:s}^k$. Let $\mathcal{A}$ be any algorithm for the $(K, B)$-secretary problem, we define its stopping time $\tau(\mathcal{A})$ as the step $t$ when it decides to return the observed candidate. We will often drop the explicit dependency on $\mathcal{A}$ and write $\tau$ when no ambiguity is involved. We will say that $\mathcal{A}$ succeeded if the selected candidate $x_\tau$ is the best among all the candidates $\{x_1, \ldots, x_N\}$. Let us also define, for any step $t \geq 1$, the random variables

$$r_t = \sum_{t'=1}^{t} \mathbb{1}(x_t \leq x_{t'}, g_t = g_{t'}) \quad \text{and} \quad R_t = \sum_{t'=1}^{t} \mathbb{1}(x_t \leq x_{t'}) \ . \tag{1}$$

Both random variables have natural interpretations: given a candidate at time $t$, $r_t$ is its *in-group rank* up to time $t$, while $R_t$ is its *overall rank* up to time $t$. Note that the actual values of $x_t$ do not play a

role in the secretary problem and we can restrict ourselves to the observations $r_t, g_t, R_t$. While the first two random variables are always available at the beginning of round $t$, the third one can be only acquired utilizing the available budget. At each step $t \in [N]$, the decision-maker observes $r_t, g_t$ and can perform one of the following three actions:

1. `skip`: reject $x_t$ and move to the next one;
2. `stop`: select $x_t$;
3. `compare`: if the comparison budget is not exhausted, use a comparison to determine if $(R_t = 1)$—compare the candidate $x_t$ to the best already seen candidates in the other groups;

Furthermore, if a comparison has been used at time $t$, the algorithm has to perform `stop` or `skip` afterward. We denote respectively by $a_{t,1}$ and $a_{t,2}$ the first and second action made by the algorithm at step $t$. Let us also define $g_t^*$ the group to which the best candidate observed until step $t$ belongs,

$$g_t^* = \underset{k \in [K]}{\operatorname{argmax}} \{\max G_t^k\} \quad \forall t \in [N] \,,$$

and $B_t$ as the budget available for $\mathcal{A}$ at step $t$

$$B_1 = B \quad \text{and} \quad B_t = B_{t-1} - \mathbb{1}(a_{t,1} = \texttt{compare}) \quad \forall t \in [N] \,.$$

In the presence of a non-zero budget, the first time when $\mathcal{A}$ decides to make a comparison will be a key parameter in our analysis of the success probability. We denote it by $\rho_1(\mathcal{A})$,

$$\rho_1(\mathcal{A}) = \min\{t \in [N] : a_{t,1} = \texttt{compare}\} \,.$$

As with the stopping time, when there is no ambiguity about $\mathcal{A}$, we simply write $\rho_1$.

**Remark 2.1.** *Although we formalized the problem using the variables $r_t$ and $R_t$, the only information needed at any step $t$ is $\mathbb{1}(r_t = 1)$ and $\mathbb{1}(R_t = 1)$, i.e we only need to know if the candidate is the best seen so far, in its own group and overall. In practice, $\mathbb{1}(r_t = 1)$ can be observed by comparing $x_t$ to the best candidate up to $t - 1$ belonging to $G^{g_t}$, and if this is the case then $\mathbb{1}(R_t = 1)$ can be observed by comparing $x_t$ to the best candidate in the other group.*

## 3 Dynamic threshold algorithm for $K$ groups

In this section, we introduce a general family of Dynamic-threshold (DT) algorithms. A DT algorithm for the $(K, B)$-secretary problem is defined by a finite doubly-indexed sequence $(\alpha_{k,b})_{k \in [K], b \leq B}$ of real numbers in $[0, 1]$, which determines the *acceptance* thresholds based on the group of the observed candidate and the available budget. During a run of the algorithm, the thresholds used for each group dynamically change depending on the evolution of the available budget. We denote this algorithm by $\mathcal{A}^B\big((\alpha_{k,b})_{k \in [K], b \leq B}\big)$.

Upon the arrival of a new candidate $x_t$, the algorithm observes its group $g_t = k \in [K]$ and its in-group rank $r_t$, and has an available budget of $B_t = b \geq 0$. If $t/N < \alpha_{k,b}$ or $r_t = 0$, the candidate is immediately rejected. Otherwise, if $t/N \geq \alpha_{k,b}$ and $r_t = 1$, then the candidate is selected if $b = 0$; and if the budget is not yet exhausted ($b > 0$), then the algorithm pays a unit cost to observe the variable $\mathbb{1}(R_t = 1)$. If this variable is 1, indicating a favorable comparison, the candidate is selected; otherwise, it is rejected. A formal description is given in Algorithm 1, and a visual representation for the case of three groups is provided in Figure 1.

### 3.1 Single-threshold algorithm for $K$ groups

In this section, we focus on the single-threshold algorithm, a specific case within the family of DT algorithms, where all thresholds are identical across groups and budgets. Initially, the algorithm rejects all candidates until step $T - 1$, where $T \in [N]$ is a fixed threshold. Upon encountering a new candidate that is the best within its group, if no budget remains, the candidate is selected. Alternatively, if there is still a budget available, the algorithm utilizes it to determine if the current candidate is the best among all groups. If that is the case, the candidate is then selected. We denote by $\mathcal{A}_T^B$ the single-threshold algorithm with threshold $T$ and budget $B$. We demonstrate that this algorithm has an asymptotic success probability converging very rapidly to the upper bound of $1/e$.

In this first lemma, we prove a recursion formula on the success probability of the single-threshold algorithm, with a threshold $T = \lfloor \alpha N \rfloor$ for some $\alpha \in [0, 1]$.

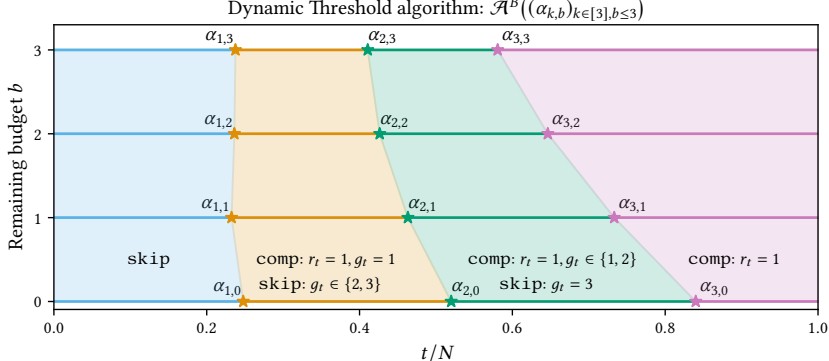

Figure 1: Schematic description of a DT algorithm in the case of 3 groups

---

**Algorithm 1:** Dynamic-Threshold algorithm $\mathcal{A}^B\big((\alpha_{k,b})_{k\in[K],b\leq B}\big)$

---

**Input:** Available budget $B$, thresholds $(\alpha_{k,b})_{k\in[K],b\leq B}$
**Initialization:** $b = B$

1 **for** $t = 1, \ldots, N$ **do**
2     Receive new observation: $(r_t, g_t)$
3     **if** $t \geq \lfloor \alpha_{g_t,b}N \rfloor$ and $r_t = 1$ **then**                 // compare in-group
4        **if** $b > 0$ **then**                                      // check budget
5           Update budget: $b \leftarrow b - 1$
6           **if** $R_t = 1$ **then** Return: $t$              // compare inter-group
7        **else** Return: $t$

---

**Lemma 3.1.** *The success probability of the single threshold algorithm $\mathcal{A}_T^B$ with threshold $T = \lfloor \alpha N \rfloor$ and budget $B \geq 0$ satisfies the recursion formula*

$$\mathbf{P}(\mathcal{A}_T^B \text{ succeeds}) = \frac{\alpha - \alpha^K}{K - 1} + \mathbb{1}(B \geq 0)(K - 1)\sum_{t=T}^{N} \frac{T^K}{t^{K+1}} \mathbf{P}(\mathcal{A}_{t+1}^{B-1} \text{ succeeds}) + O\left(\sqrt{\frac{\log N}{N}}\right).$$

The proof hinges on analyzing the behavior of the algorithm following the first comparison. After that comparison, the algorithm halts if $R_t = 1$, and the success probability can be computed in that case. Otherwise, if $R_t \neq 1$, the candidate is rejected, and the algorithm transitions to a new state at step $t + 1$, where the available budget reduces to $B - 1$. Its success probability becomes precisely that of algorithm $\mathcal{A}_{t+1}^{B-1}$, with budget $B - 1$ and threshold $t + 1$.

The recursion outlined in Lemma 3.1 can be used to calculate the asymptotic success probability of the single-threshold algorithm $\mathcal{A}_{\lfloor \alpha N \rfloor}^B$ as the number of candidates $N$ approaches infinity.

**Theorem 3.2.** *The asymptotic success probability of the single threshold algorithm $\mathcal{A}_T^B$ with threshold $T = \lfloor \alpha N \rfloor$ and budget $B \geq 0$ is*

$$\lim_{N \to \infty} \mathbf{P}(\mathcal{A}_{\lfloor \alpha N \rfloor}^B \text{ succeeds}) = \frac{\alpha^K}{K - 1} \sum_{b=0}^{B} \left(\frac{1}{\alpha^{K-1}} - \sum_{\ell=0}^{b} \frac{\log(1/\alpha^{K-1})^\ell}{\ell!}\right).$$

*In particular,*

$$\lim_{B \to \infty} \lim_{N \to \infty} \mathbf{P}(\mathcal{A}_{\lfloor \alpha N \rfloor}^B \text{ succeeds}) = \alpha \log(1/\alpha).$$

Note that, for $B = \infty$, the asymptotic success probability in the previous theorem corresponds to the success probability of the algorithm with a threshold $\lfloor \alpha N \rfloor$ in the secretary problem. Indeed, with an unlimited budget, the decision-maker can assess at each step whether the current candidate is the best so far, and the problem becomes equivalent to the classical secretary problem.

**Alternative comparison model.** In the alternative comparison model presented in Section 2, the single threshold algorithm $\mathcal{A}_T^B$ can be adapted to guarantee the same success probability at the cost of $K - 1$ additional comparisons. After the first $T$ candidates are rejected, $K - 1$ comparisons are made between the maximums from each group to identify the best candidate so far. The algorithm then keeps track of the best candidate: whenever a new candidate is the best in their group, they are compared to the current best candidate using a single comparison, and the latter is updated accordingly. This approach enjoys the same guarantees as in Theorem 3.2, but with a budget of $K + B - 1$ instead of $B$.

The next corollary measures how the success probability of the single threshold algorithm, in the setting with $K$ groups, converges to $1/e$ as the budget increases.

**Corollary 3.2.1.** *The success probability of the single-threshold algorithm with threshold $T = \lfloor N/e \rfloor$ and budget $B \geq 0$ satisfies*

$$\lim_{N \to \infty} \mathbf{P}(\mathcal{A}_{\lfloor N/e \rfloor}^B \text{ succeeds}) \geq \frac{1}{e} \left( 1 - \frac{(K-1)^{B+1}}{(B+1)!} \right) .$$

*In particular, for all $\varepsilon > 0$, if $K \leq 1 + \frac{B+1}{e}(e\varepsilon)^{\frac{1}{B+1}}$, then $\lim_N \mathbf{P}(\mathcal{A}_{\lfloor N/e \rfloor}^B \text{ succeeds}) \geq (1 - \varepsilon)/e$.*

This corollary proves that the success probability of $\mathcal{A}_{\lfloor N/e \rfloor}^B$ converges very rapidly to the upper bound $1/e$ as $B$ increases. However, the convergence becomes slower when $K$ is larger.

Surprisingly, the asymptotic success probability of $\mathcal{A}_{\lfloor N/e \rfloor}^B$ is not influenced by the proportions $(\lambda_k)_{k \in [K]}$, but only by the number of groups $K$. This means that the algorithm does not benefit from the cases where there is a majority group $G^k$ with $\lambda_k$ very close to 1, which would make the problem easier. Indeed, it is always possible to achieve a success probability of $\max_{k \in [K]} \lambda_k/e$ by rejecting all the candidates not belonging to the majority group $G^{k^*}$, and using the classical $1/e$-rule counting only elements of $G^{k^*}$. This algorithm can be combined with ours by always running the one with the highest success probability, depending on the available budget, the number of groups, and the group proportions. The resulting algorithm has a success probability that converges to the upper bound $1/e$ both when $B$ increases and when $\max_k \lambda_k$ converges to 1. Nonetheless, due to the very fast convergence of the success probability of the single threshold algorithm to $1/e$, the improvement brought by having a majority group is only marginal when the budget is sufficient.

As a consequence, the single threshold algorithm surprisingly constitutes a very efficient solution to the problem even with moderate values of the budget. Computing the optimal thresholds remains however an intriguing question, which we explore in the following sections in the case of two groups.

## 4 The case of two groups

In this section, we delve into the particular case of two groups, and we demonstrate how leveraging different thresholds for each group can enhance the success probability. Let $\lambda \in (0, 1)$ represent the probability of belonging to group $G^1$, and $1 - \lambda$ the probability of belonging to group $G^2$. We examine the success probability of Algorithm $\mathcal{A}^B(\alpha, \beta)$, with threshold $\lfloor \alpha N \rfloor$ for group $G^1$ and $\lfloor \beta N \rfloor$ for group $G^2$, having a budget of $B$ comparisons. This algorithm is a specific instance of the DT family, wherein the thresholds depend only on the group, and not on the available budget. We call it a *static double-threshold* algorithm.

We assume without loss of generality that $\alpha \leq \beta$, and we denote by $\mathcal{C}_N$ the event

$$(\mathcal{C}_N) : \quad \forall t \geq 1 : \max(||G_t^1| - \lambda t| , ||G_t^2| - (1-\lambda)t|) \leq 4\sqrt{t \log N} .$$

This event provides control over the group sizes at each step. Lemma A.2 guarantees that $\mathcal{C}_N$ holds true with a probability at least $1 - \frac{1}{N^2}$ for $N \geq 4$. Furthermore, for all $t \in [N]$, we denote by $\mathcal{A}_t^B(\alpha, \beta)$ the algorithm with acceptance thresholds $\max\{\lfloor \alpha N \rfloor, t\}$ and $\max\{\lfloor \beta N \rfloor, t\}$ respectively for groups $G^1$ and $G^2$, and we denote by $\mathcal{U}_{N,t,k}^B$ the probability

$$\mathcal{U}_{N,t,k}^B = \mathbf{P}(\mathcal{A}_t^B(\alpha, \beta) \text{ succeeds}, g_{t-1}^* = k \mid \mathcal{C}_N) . \tag{2}$$

Similar to the analysis of the single-threshold algorithm, we establish in Lemma C.1 a recursion formula satisfied by $(\mathcal{U}_{N,t,k}^B)_{B,t,k}$, which we later utilize to derive lower bounds on the asymptotic

success probability of $\mathcal{A}^B(\alpha, \beta)$. To prove this lemma, we study the probability distribution of the occurrence time $\rho_1$ of the first comparison made by $\mathcal{A}_t^B(\alpha, \beta)$, and we examine the algorithm's success probability following it. Essentially, if $\rho_1 = s$, we can compute the probability of stopping and the corresponding success probability, and the distribution of the state of the algorithm at step $s + 1$, which yields the recursion. Using adequate concentration arguments and Lemma C.1, we show the two following results, giving explicit recursive formulas satisfied by the limit of $\mathcal{U}_{N,t,k}^B$ when $N \to \infty$, respectively for $k = 2$ and $k = 1$.

**Lemma 4.1.** *For all $B \geq 0$ and $w \in [\alpha, \beta]$, the limit $\varphi_2^B(\alpha, \beta; w) = \lim_{N \to \infty} \mathcal{U}_{N, \lfloor wN \rfloor, 2}^B$ exists, and it satisfies the following recursion*

$$
\varphi_2^B(\alpha, \beta; w) = -\lambda w \log \left((1 - \lambda)\tfrac{w}{\beta} + \lambda\right) + \frac{(1 - \lambda)\beta w^2}{(1 - \lambda)w + \lambda\beta} \sum_{b=0}^{B} \left(\frac{1}{\beta} - \sum_{\ell=0}^{b} \frac{\log(1/\beta)^\ell}{\ell!}\right)
$$

$$
+ \mathbb{1}(B > 0)\, w^2 \int_w^\beta \frac{(1 - \lambda)^2 w + \lambda(2 - \lambda)u}{((1 - \lambda)w + \lambda u)^2 u^2} \varphi_2^{B-1}(\alpha, \beta; u)du \ .
$$

*Moreover, $\mathcal{U}_{N, \lfloor wN \rfloor, 2}^B = \varphi_2^B(\alpha, \beta; w) + O\left(\sqrt{\frac{\log N}{N}}\right)$.*

**Lemma 4.2.** *For all $B \geq 0$ and $w \in [\alpha, \beta]$, the limit $\varphi_1^B(\alpha, \beta; w) = \lim_{N \to \infty} \mathcal{U}_{N, \lfloor wN \rfloor, 1}^B$ exists, and it satisfies the following recursion*

$$
\varphi_1^B(\alpha, \beta; w) = \lambda w \log \left(1 - \lambda + \lambda\tfrac{\beta}{w}\right) + \frac{\lambda w \beta^2}{(1 - \lambda)w + \lambda\beta} \sum_{b=0}^{B} \left(\frac{1}{\beta} - \sum_{\ell=0}^{b} \frac{\log(1/\beta)^\ell}{\ell!}\right)
$$

$$
+ \mathbb{1}(B > 0)\, \lambda^2 w \int_w^\beta \frac{(u - w)\varphi_2^{B-1}(\alpha, \beta; u)}{((1 - \lambda)w + \lambda u)^2 u} du \ .
$$

*Moreover, $\mathcal{U}_{N, \lfloor wN \rfloor, 1}^B = \varphi_1^B(\alpha, \beta; w) + O\left(\sqrt{\frac{\log N}{N}}\right)$.*

We deduce that the asymptotic success probability of Algorithm $\mathcal{A}_{\lfloor wN \rfloor}^B$, conditioned on the event $\mathcal{C}_N$, exists and equals $\varphi_1^B(\alpha, \beta; w) + \varphi_2^B(\alpha, \beta; w)$. Additionally, by applying Lemma A.3, we eliminate the conditioning on $\mathcal{C}_N$, thus proving the following theorem.

**Theorem 4.3.** *For all $0 < \alpha \leq \beta \leq 1$, The success probability of Algorithm $\mathcal{A}^B(\alpha, \beta)$ satisfies*

$$
\mathbf{P}(\mathcal{A}^B(\alpha, \beta) \text{ succeeds}) - O\left(\sqrt{\frac{\log N}{N}}\right)
$$

$$
= \lambda\alpha \log\left(\tfrac{\beta}{\alpha}\right) + \alpha\beta \sum_{b=0}^{B} \left(\frac{1}{\beta} - \sum_{\ell=0}^{b} \frac{\log(1/\beta)^\ell}{\ell!}\right) + \mathbb{1}(B > 0)\, \alpha \int_\alpha^\beta \frac{\varphi_2^{B-1}(\alpha, \beta; u)du}{u^2} \ ,
$$

*with $\varphi_2^B(\alpha, \beta; \cdot)$ defined in Lemma 4.1.*

It is possible to use Theorem 4.3 and 4.1 to numerically compute the success probability of $\mathcal{A}^B(\alpha, \beta)$. However, this computation is heavy due the recursion defining $\varphi_2^B(\alpha, \beta; w)$. Moreover, it is difficult to prove a closed expression, and even more to compute the optimal thresholds.

By disregarding the term containing $\varphi^2(\alpha, \beta; \cdot)$ in the theorem, we derive an analytical lower bound expressed as a function of the parameters $\lambda$, $B$, $\alpha$, and $\beta$, allowing a more effective threshold selection. In the subsequent discussion, for all $w \in (0, 1]$ and $B \geq 0$, we denote by $S^B(w)$ the following sum:

$$
S^B(w) = \sum_{b=0}^{B} \left(\frac{1}{w} - \sum_{\ell=0}^{b} \frac{\log(1/w)^\ell}{\ell!}\right) \ .
$$

**Corollary 4.3.1.** *Assume that $\lambda \geq 1/2$. Let $h^B : \beta \mapsto \min\left\{\frac{\beta}{e} \exp\left(\frac{\beta S^B(\beta)}{\lambda}\right), \beta\right\}$, and $\tilde{\alpha}_B, \tilde{\beta}_B$ the thresholds defined as $\tilde{\alpha}_B = h^B(\tilde{\beta}_B)$, and $\tilde{\beta}_B$ minimizing the mapping*

$$
\beta \in [0, 1] \mapsto \lambda h(\beta) \log\left(\tfrac{\beta}{h^B(\beta)}\right) + h^B(\beta)\beta S^B(\beta) \ ,
$$

*then the success probability of $\mathcal{A}^B(\tilde{\alpha}_B, \tilde{\beta}_B)$ satisfies*

$$
\lim_{N \to \infty} \mathbf{P}(\mathcal{A}^B(\tilde{\alpha}_B, \tilde{\beta}_B) \text{ succeeds}) \geq \frac{1}{e} - \min\left\{\frac{1}{e(B + 1)!}, (\tfrac{4}{e} - 1)\lambda(1 - \lambda)\right\} \ .
$$

Therefore, in contrast to the single-threshold algorithm, the asymptotic success probability of $\mathcal{A}^B(\tilde{\alpha}_B, \tilde{\beta}_B)$ approaches $1/e$ both when the budget increases and when $\lambda$ approaches 0 or 1.

## 4.1 Optimal memory-less algorithm for two groups

In the following, an algorithm is called *memory-less* if its actions at any step $t \in [N]$ depend only on the current observations $r_t, g_t, \mathbb{1}(R_t = 1)$, the available budget $B_t$, and the cardinals $(|G_{t-1}^k|)_{k \in [K]}$.

We use in this section a dynamic programming approach to determine the optimal memory-less algorithm, which we denote by $\mathcal{A}_*$.
Unlike previous sections, our analysis here is not asymptotic. By meticulously examining how various variables, including the precise number of candidates observed in each group, influence the success probability of $\mathcal{A}_*$, we rigorously analyze its state transitions and corresponding success probabilities to determine optimal actions at each step. A full description and analysis of the optimal memory-less algorithm can be found in Section D. Here, we illustrate its actions through Figure 2.

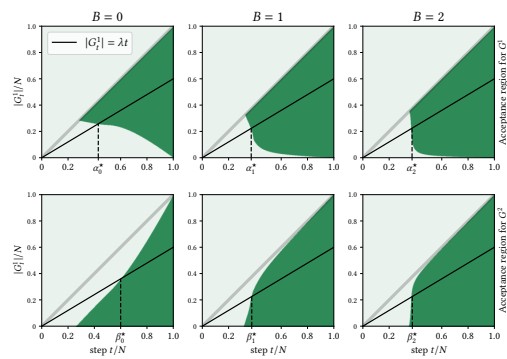

Figure 2: Acceptance region of $\mathcal{A}_*$

**Computing optimal thresholds for the DT algorithm.** Upon observing $(r_t, g_t)$, $\mathcal{A}_*$ makes a decision to accept or reject, where acceptance means stop if $B = 0$ and compare otherwise, depending on $t, |G_t^1|, B, g_t$. Figure 2 shows its acceptance region (dark green), with $N = 500$, $\lambda = 0.7$, for $B \in \{0, 1, 2\}$ and for all possible values of $t \in [N]$, $|G_t^1| \leq t$, and $g_t \in \{1, 2\}$. The x- and y-axes display respectively the step $t$ and possible group cardinal $|G_t^1|$, which implies $|G_t^2|$, up to time $t$. The latter follows a binomial distribution with parameters $(\lambda, t)$, which tightly concentrates around its mean $|G_t^1| \approx \lambda t$ (and $|G_t^2| \approx (1 - \lambda)t$) even for moderate values of $t$. Consequently, when $N$ is large, $|G_t^1| \approx \lambda t$, and the acceptance region is solely defined by a threshold at the intersection of the acceptance region and the line $|G_t^1| \approx \lambda t$. This observation implies that $\mathcal{A}_*$ behaves as an instance of DT algorithms when the number of candidates is large. The corresponding thresholds, which we denote by $(\alpha_b^\star, \beta_b^\star)_{b \leq B}$, are necessarily optimal, and can be estimated as the intersection of the acceptance region for $G^k$ and the line $(t, \lambda t)$ for $k \in \{1, 2\}$.

**Alternative comparison model.** In the particular case of two groups, both comparison models introduced in Section 2 are equivalent, as freely comparing a candidate with the best in their group and then making one costly comparison with the best candidate from the other group is sufficient to determine if they are the best so far. Therefore, all the results of the current section regarding the static double-threshold algorithm and optimal memory-less algorithm remain true in the alternative comparison model.

## 5 Numerical experiments

In this section, we confirm our theoretical findings via numerical experiments, and we give further insight regarding the behavior of the algorithms we presented and how they compare to each other. In all the empirical experiments of this section, each point is computed over $10^6$ independent trials. The code used for the experiments is available at github.com/Ziyad-Benomar/Addressing-bias-in-online-selection-with-limited-budget-of-comparisons.

### 5.1 Single-threshold algorithm

Using Theorem 3.2, the optimal threshold, for the single-threshold algorithm, can be computed numerically for fixed $K$ and $B$. Figure 3 illustrates the optimal threshold and the corresponding success probability for $B \in \{0, \ldots, 30\}$ and $K \in \{2, 10, 25, 50\}$. For any $K \geq 2$, as the budget grows to infinity, the problem becomes akin to the standard secretary problem, leading the optimal

threshold to converge to $1/e$. However, as discussed in Corollary 3.2.1, the convergence is slower when the number of groups $K$ is higher.

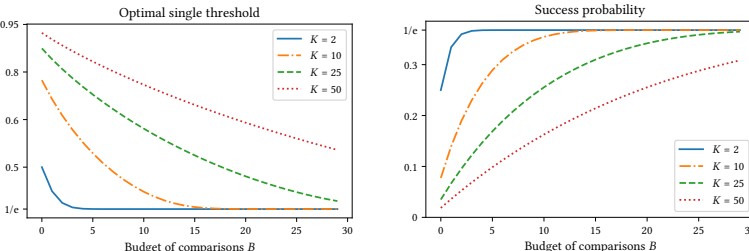

Figure 3: Single threshold algorithm: optimal threshold and corresponding success probability

Moreover, Theorem 3.2 reveals that the asymptotic success probability is independent of the probabilities of belonging to each group, and it is equal to a value smaller than $1/e$. This indicates a discontinuity of the success probability at the extreme points of the polygon defining the possible values of $(\lambda_k)_{k \in [K]}$. Figure 4 illustrates this behavior for the case of two groups, with $N = 500$ candidates, and $B \in \{0, 1, 2\}$. On the other hand, while our theoretical results study asymptotic success probabilities, they do not comprehend how the performance of the algorithms varies with the number of candidates. Figure 5 shows that the success probability is better when the number of candidates is small, and it decreases to match the asymptotic expression when $N \to \infty$, represented with dotted lines, for $K \in \{2, 3, 4\}$, with $B = 3$.

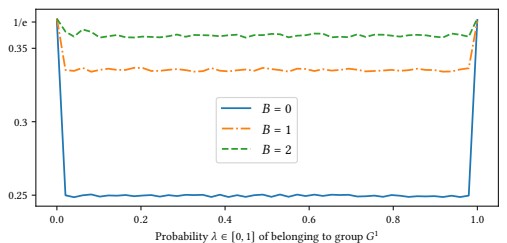

Figure 4: Single threshold: success probability for 2 groups, with $N = 500$ and $\lambda \in [0, 1]$

Figure 5: Convergence to the asymptotic success probability, with $\lambda_k = 1/K$ for all $k \in [K]$

## 5.2 The case of two groups

In the case of two groups, Figure 3 shows that, even with a very limited budget, the single-threshold algorithm has a success probability almost indistinguishable from the upper bound $1/e$. Consequently, in the remaining experiments in the two-group scenario, we restrict ourselves to small budgets $B \leq 3$.

Theorem D.4 shows a recursive formula for computing the success probability of the optimal memory-less algorithm $\mathcal{A}_*$ for all $N \geq 1$, $B \geq 0$, and $\lambda \in (0, 1)$. Figure 6 displays this success probability, in solid lines, for $N = 500$ and $B \in \{0, 1, 2\}$, along with the success probability of the static double-threshold algorithm $\mathcal{A}^B(\tilde{\alpha}_B, \tilde{\beta}_B)$ in dotted lines, where $\tilde{\alpha}_B, \tilde{\beta}_B$ are defined in Corollary 4.3.1. The figure demonstrates that for $B = 0$, or $\lambda$ close to $0.5$, Algorithm $\mathcal{A}^B(\tilde{\alpha}_B, \tilde{\beta}_B)$ matches the performance of $\mathcal{A}_*$, despite having a much simpler structure.

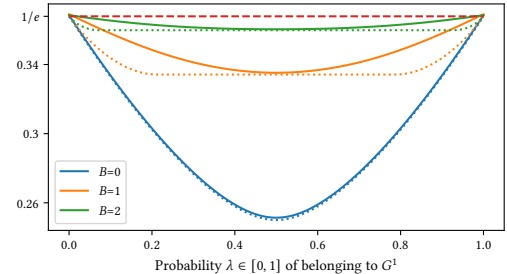

Figure 6: Success probability of $\mathcal{A}_*$, and the lower bound of Corollary 4.3.1

For $\lambda = 0.5$, both groups have symmetric roles, and the optimal thresholds $\tilde{\alpha}_B$ and $\tilde{\beta}_B$ to choose in the static-threshold algorithm are identical. Hence, the success probability of $\mathcal{A}^B(\tilde{\alpha}_B, \tilde{\beta}_B)$ for $\lambda = 0.5$ is exactly that of the single-threshold algorithm, which is independent of $\lambda$ (Theorem 3.2). We deduce from this observation and Figure 6 that having different thresholds for each group yields a substantial improvement over the single-threshold algorithm when $\lambda$ is close to 0 or 1.

Finally, to emphasize that the dynamic programming algorithm $\mathcal{A}_*$ is equivalent to an instance of DT algorithm for large $N$, Figure 7 compares the empirical success probabilities of $\mathcal{A}_*$ (dotted lines) and the DT algorithm (solid lines) with thresholds $(\alpha_B^\star, \beta_B^\star)_{B \geq 0}$, computed as explained in Section 4.1.

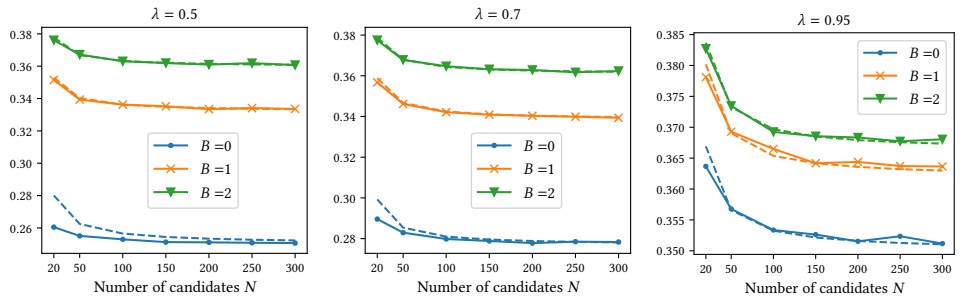

Figure 7: Empirical success probabilities of $\mathcal{A}_*$ and the DT algorithm with optimal thresholds

For $\lambda \in \{0.5, 0.7, 0.95\}$, the figure confirms that, despite the intricate structure of the optimal memory-less algorithm, it does not surpass the performance of the DT algorithm with optimal thresholds when $N$ is large. Nonetheless, the analysis of the optimal memory-less algorithm is what enables the numerical computation of the optimal thresholds, as explained previously. Figure 8 shows the optimal thresholds $\alpha_b^\star, \beta_b^\star$ for all $\lambda \in [0.5, 1]$ and $B \in \{0, 1, 2\}$.

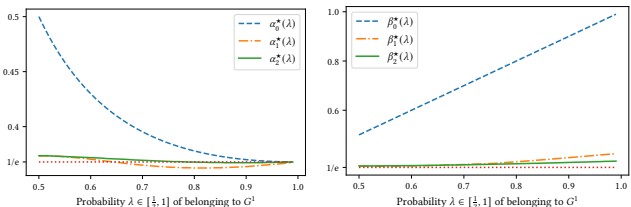

Figure 8: $(\alpha_b^\star, \beta_b^\star)$ as functions of $\lambda$.

These thresholds are continuous functions of $\lambda$, both converging to $1/e$ very rapidly as the budget increases. Indeed, for $B \geq 1$, they both become very close to $1/e$, as for $B = 0$, the optimal thresholds are exactly equal to $\alpha_0^\star = \lambda \exp(\frac{1}{\lambda} - 2)$ and $\beta_0^\star = \lambda$, which correspond to the optimal thresholds described in Corollary 4.3.1 for $B = 0$ (See the proof of the corollary).

# 6 Conclusion and future work

This paper explores more realistic online selection processes, wherein nuanced factors such as imperfect comparisons and optimal budget utilization are considered. We studied a partially ordered secretary problem, wherein a constrained budget of comparisons is allowed. Our findings encompass both asymptotic and non-asymptotic analyses. Specifically, we explore the asymptotic behavior of the single threshold algorithm with $K$ groups, demonstrating its high efficiency with a non-zero budget. Furthermore, in the context of two groups, we study the success probability of static double-threshold algorithms, and we present a non-asymptotic optimal memoryless algorithm. Through numerical experimentation, we demonstrate that this algorithm behaves as a DT algorithm, and, leveraging this insight, we show how to numerically compute the optimal DT thresholds. However, a limitation of the paper is that optimal thresholds are only computed in the case of two groups. Future investigation could explore methods for numerically or analytically characterizing optimal DT thresholds with an arbitrary number of groups $K$.

## Acknowledgements

This research was supported in part by the French National Research Agency (ANR) in the framework of the PEPR IA FOUNDRY project (ANR-23-PEIA-0003) and through the grant DOOM ANR-23-CE23-0002. It was also funded by the European Union (ERC, Ocean, 101071601). Views and opinions expressed are however those of the author(s) only and do not necessarily reflect those of the European Union or the European Research Council Executive Agency. Neither the European Union nor the granting authority can be held responsible for them.

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

# A  Preliminaries

With the assumption that the group membership of each candidate is a random variable, to compute the asymptotic success probabilities of the algorithms presented in the paper, it is necessary to use concentration inequalities to estimate the number of candidates in each group. We use Lemma 3 from Jamieson et al. [2014] to prove the following.

**Lemma A.1** (Jamieson et al. [2014])**.** *Let $(X_t)_{t \geq 1}$ be i.i.d. Bernoulli random variables, $N$ a positive integer, and $m > 0$ satisfying $N^m > 8$, then it holds with a probability of at least $1 - \frac{25}{N^{2m}}$ that*

$$\forall t \geq 1 : \sum_{s=1}^{t} (X_s - \mathbf{E}[X_s]) \leq 2\sqrt{(m+1)t \log N}$$

*Proof.* Bernoulli random variables are sub-Gaussian with scale parameter $1/2$, hence Lemma 3 from Jamieson et al. [2014] with $\varepsilon = 1$ guarantees that, with a probability of at least $1 - 3(\frac{\delta}{\log 2})^2$, the following holds

$$\forall t \geq 1 : \sum_{s=1}^{t} (X_s - \mathbf{E}[X_s]) \leq 2\sqrt{t \log \left( \frac{\log(2t)}{\delta} \right)} \ .$$

Consider positive integers $T \leq N$, $m \geq 1$ such that $N^m > 8$, and $\delta = 2/N^m$. Then for all $t \in \{T, \dots, N\}$

$$\frac{\log(2t)}{\delta} \leq \frac{2t}{\delta} = N^m t \leq N^{m+1}$$

and we deduce that

$$\mathbf{P}\Big(\forall t \geq 1 : \sum_{s=1}^{t} (X_s - \mathbf{E}[X_s]) \leq 2\sqrt{(m+1)t \log N}\Big)$$

$$\geq \mathbf{P}\left(\forall t \geq 1 : \sum_{s=1}^{t} (X_s - \mathbf{E}[X_s]) \leq 2\sqrt{t \log \left( \frac{\log(2t)}{2/N^m} \right)}\right)$$

$$\geq 1 - \frac{3(2/\log 2)^2}{N^{2m}}$$

$$\geq 1 - \frac{25}{N^{2m}} \ .$$

$\square$

In the context of the $K$-group secretary problem, adequately using the previous Lemma and using union bounds yields a concentration inequality on the number of candidates belonging to each group.

**Lemma A.2.** *Let $N \geq \max(4, K)$, and consider the following event*

$$\forall k \in [K], \forall t \geq 1 : ||G_t^k| - \lambda_k t| \leq 4\sqrt{t \log N} \ ,$$

*which we denote by $\mathcal{C}_N$. Then it holds that $\mathbf{P}(\mathcal{C}_N) \geq 1 - \frac{1}{N^2}$.*

*Proof.* The previous Lemma with $N \geq 4$, $m = 3$ and respectively with the Bernoulli random variables $X_{t,k,0} = \mathbb{1}(g_t = k)$ and $X_{t,k,1} = 1 - \mathbb{1}(g_t = k)$, gives for all $i \in \{0, 1\}$ and $k \in [K]$ that

$$\mathbf{P}\left(\forall t \geq 1 : (-1)^i (|G_t^k| - \lambda_k t) \leq 4\sqrt{t \log N}\right) \geq 1 - \frac{25}{N^6} \ ,$$

and we obtain by a union bound that $\mathbf{P}(\mathcal{C}_N) \geq 1 - \frac{50K}{N^6}$. Given that $N \geq \max(4, K)$, it follows that

$$\mathbf{P}(\mathcal{C}_N) \geq 1 - \frac{K}{N} \cdot \frac{50}{N^3} \cdot \frac{1}{N^2} \geq 1 - \frac{1}{N^2} \ .$$

$\square$

**Lemma A.3.** *Let $\mathcal{E}$ be any event, not necessarily independent of $\mathcal{C}_N$ (defined in Lemma A.2), then*

$$\mathbf{P}(\mathcal{E} \mid \mathcal{C}_N) = \mathbf{P}(\mathcal{E}) + O(1/N^2) \ .$$

*Proof.* We have by Lemma A.2 that

$$\mathbf{P}(\mathcal{E} \mid \mathcal{C}_N) = \frac{\mathbf{P}(\mathcal{E} \cap \mathcal{C}_N)}{\mathbf{P}(\mathcal{C}_N)} \leq \frac{\mathbf{P}(\mathcal{E})}{1 - \frac{1}{N^2}} \leq \left(1 + \frac{2}{N^2}\right)\mathbf{P}(\mathcal{E}) \leq \mathbf{P}(\mathcal{E}) + \frac{2}{N^2}.$$

Using the same inequality with the complementary event $\mathcal{E}^c$ of $\mathcal{E}$ gives

$$\mathbf{P}(\mathcal{E} \mid \mathcal{C}_N) = 1 - \mathbf{P}(\mathcal{E}^c \mid \mathcal{C}_N) \geq 1 - \left(\mathbf{P}(\mathcal{E}^c) + \frac{2}{N^2}\right) = \mathbf{P}(\mathcal{E}) - \frac{2}{N^2},$$

which concludes the proof. $\qquad\square$

**Lemma A.4.** *For any $x > 0$ and for any $B \geq 1$ we have*

$$0 \leq xe^x - \sum_{b=0}^{B}\left(e^x - \sum_{\ell=0}^{b}\frac{x^\ell}{\ell!}\right) \leq e^x\frac{x^{B+2}}{(B+1)!}.$$

*Proof.* Let $x > 0$ and $B \geq 1$. First, observe that

$$\sum_{b=0}^{\infty}\left(e^x - \sum_{\ell=0}^{b}\frac{x^\ell}{\ell!}\right) = \sum_{b=0}^{\infty}\sum_{\ell=b+1}^{\infty}\frac{x^\ell}{\ell!} = \sum_{\ell=1}^{\infty}\sum_{b=0}^{\ell-1}\frac{x^\ell}{\ell!} = \sum_{\ell=1}^{\infty}\frac{x^\ell}{(\ell-1)!} = x\sum_{\ell=0}^{\infty}\frac{x^\ell}{\ell!} = xe^x,$$

therefore

$$\sum_{b=0}^{B}\left(e^x - \sum_{\ell=0}^{b}\frac{x^\ell}{\ell!}\right) \leq xe^x,$$

and we have

$$xe^x - \sum_{b=0}^{B}\left(e^x - \sum_{\ell=0}^{b}\frac{x^\ell}{\ell!}\right) = \sum_{b=B+1}^{\infty}\sum_{\ell=b+1}^{\infty}\frac{x^\ell}{\ell!} = \sum_{\ell=B+2}^{\infty}\sum_{b=B+1}^{\ell-1}\frac{x^\ell}{\ell!} = \sum_{\ell=B+2}^{\infty}(\ell - B - 1)\frac{x^\ell}{\ell!}$$

$$\leq \sum_{\ell=B+2}^{\infty}\frac{x^\ell}{(\ell-1)!} \leq x\sum_{\ell=B+1}^{\infty}\frac{x^\ell}{\ell!} \leq e^x\frac{x^{B+2}}{(B+1)!},$$

where we used for the last step the classical inequality

$$\sum_{\ell=B+1}^{\infty}\frac{x^\ell}{\ell!} = e^x - \sum_{\ell=0}^{B}\frac{x^\ell}{\ell!} \leq e^x\frac{x^{B+1}}{(B+1)!}.$$

$\qquad\square$

# B  Single-Threshold algorithms for $K$ groups

## B.1  Proof of Lemma 3.1

*Proof.* We consider that $B \geq 1$. The case $B = 0$ is treated separately at the end of the proof. Let $\mathcal{C}_N$ the event $(\forall k \in [K], \forall t \geq 1 : ||G_t^k| - \lambda_k t| \leq 4\sqrt{t \log N})$. It holds that

$$\mathbf{P}(\mathcal{A}_T^B \text{ succeeds} \mid \mathcal{C}_N) = \sum_{t=T}^{N}\sum_{k=1}^{K}\sum_{\ell=1}^{K}\mathbf{P}(\mathcal{A}_T^B \text{ succeeds}, \rho_1 = t, g_t = \ell, g_{T-1}^* = k \mid \mathcal{C}_N)$$

$$= \sum_{t=T}^{N}\sum_{k=1}^{K}\sum_{\ell=1}^{K}\mathbf{P}(\mathcal{A}_T^B \text{ succeeds}, R_t = 1, \rho_1 = t, g_t = \ell, g_{T-1}^* = k \mid \mathcal{C}_N)$$

$$+ \sum_{t=T}^{N}\sum_{k=1}^{K}\sum_{\ell=1}^{K}\mathbf{P}(\mathcal{A}_T^B \text{ succeeds}, R_t \neq 1, \rho_1 = t, g_t = \ell, g_{T-1}^* = k \mid \mathcal{C}_N).$$

$$(3)$$

In the following, we will estimate the terms in both sums. We recall that we consider $K$ and $(\lambda_k)_{k\in[K]}$ to be constants, and $T = \alpha N + o(1)$, with $\alpha$ also a constant. All the $O$ terms appearing in the proof are independent of $t$, as $T \le t \le N$, they only depend on $N$, $T/N = \alpha + o(1)$, and the constant parameters.

For $t \in \{T, \ldots, N\}$, if $\rho_1 = t$ and $R_t = 1$, then the Algorithms stops on $x_t$, hence its succeeds if and only if $x_t = x_{\max}$. The event $x_t = x_{\max}$ is independent of the group membership of the candidates, thus independent of $\mathcal{C}_N$, and its probability is $1/N$. The event $g_t = \ell$, however, is not independent of $\mathcal{C}_N$, but Lemma A.3 gives that $\mathbf{P}(g_t = \ell \mid \mathcal{C}_N) = \mathbf{P}(g_t = \ell) + O(1/N^2) = \lambda_\ell + O(1/N^2)$. Therefore, it holds for all $k, \ell \in [K]$ and $t \in \{T, \ldots, N\}$ that

$$
\begin{aligned}
\mathbf{P}(\mathcal{A}_T^B \text{ succeeds}, R_t = 1, &\rho_1 = t, g_t = \ell, g_{T-1}^* = k \mid \mathcal{C}_N) \\
&= \mathbf{P}(x_t = x_{\max}, \rho_1 \ge t, g_t = \ell, g_{T-1}^* = k \mid \mathcal{C}_N) \\
&= \mathbf{P}(x_t = x_{\max})\mathbf{P}(g_t = \ell \mid \mathcal{C}_N)\mathbf{P}(\rho_1 \ge t, g_{T-1}^* = k \mid \mathcal{C}_N) \\
&= \left( \frac{\lambda_\ell}{N} + O(1/N^3) \right) \mathbf{P}(\rho_1 \ge t, g_{T-1}^* = k \mid \mathcal{C}_N) .
\end{aligned}
$$

Note that, for the single-threshold algorithm, we have the equivalence $\rho_1 = t \iff \rho_1 \ge t$ and $r_t = 1$. The event $\rho_1 \ge t$ happens if and only if no candidate $x_s$ for $s \in \{T, \ldots, t-1\}$ in any group $m \in [K]$ exceeds the best candidate seen up to time $T - 1$ in the same group:

$$
\forall t \ge T : (\rho_1 \ge t) \iff (\forall m \in [K] : \max G_{T:t-1}^m < \max G_{T-1}^m) ,
$$

with the convention $\max \emptyset = -\infty$. Consequently, if $\rho_1 \ge t$, then $g_{T-1}^* = k$ means that the best candidate in all groups until time $t - 1$ belongs to group $G_{T-1}^k$. Using that $T = \Theta(N)$, this yields

$$
\begin{aligned}
\mathbf{P}(\mathcal{A}_T^B &\text{ succeeds}, R_t = 1, \rho_1 = t, g_t = \ell, g_{T-1}^* = k \mid \mathcal{C}_N) \\
&= \left( \frac{\lambda_\ell}{N} + O(1/N^3) \right) \mathbf{P}(\rho_1 \ge t \text{ and } \max x_{1:t-1} \in G_{T-1}^k \mid \mathcal{C}_N) \\
&= \left( \frac{\lambda_\ell}{N} + O(1/N^3) \right) \mathbf{P}(\max x_{1:t-1} \in G_{T-1}^k \mid \mathcal{C}_N)\mathbf{P}(\rho_1 \ge t \mid \max x_{1:t-1} \in G_{T-1}^k, \mathcal{C}_N) \\
&= \left( \frac{\lambda_\ell}{N} + O(1/N^3) \right) \left( \frac{\lambda_k(T-1)}{t-1} + O(1/N^2) \right) \mathbf{P}(\rho_1 \ge t \mid \max x_{1:t-1} \in G_{T-1}^k, \mathcal{C}_N) \\
&= \left( \frac{\lambda_\ell\lambda_k T}{Nt} + O(1/N^3) \right) \mathbf{P}(\forall m \in [K] \setminus \{k\} : \max G_{T:t-1}^m < \max G_{T-1}^m \mid \mathcal{C}_N) \\
&= \left( \frac{\lambda_\ell\lambda_k T}{Nt} + O(1/N^3) \right) \prod_{m \ne k} \mathbf{E}\left[ \frac{|G_{T-1}^k|}{|G_{t-1}^k|} \,\Big|\, \mathcal{C}_N \right] \\
&= \left( \frac{\lambda_\ell\lambda_k T}{Nt} + O(1/N^3) \right) \prod_{m \ne k} \mathbf{E}\left[ \frac{\lambda_m T + O(\sqrt{N \log N})}{\lambda_m t + O(\sqrt{N \log N})} \,\Big|\, \mathcal{C}_N \right] \\
&= \left( \frac{\lambda_\ell\lambda_k T}{Nt} + O(1/N^3) \right) \prod_{m \ne k} \left( \frac{T}{t} + O\left( \sqrt{\frac{\log N}{N}} \right) \right) \\
&= \left( \frac{\lambda_\ell\lambda_k T}{Nt} + O(1/N^3) \right) \left( \frac{T^{K-1}}{t^{K-1}} + O\left( \sqrt{\frac{\log N}{N}} \right) \right) \\
&= \frac{\lambda_\ell\lambda_k}{N}(T/t)^K + O\left( \sqrt{\frac{\log N}{N^3}} \right) .
\end{aligned} \tag{4}
$$

On the other hand, regarding the terms of the second sum in (3), if $\rho_1 = t$ but $R_t \ne 1$, the Algorithm uses a comparison to observe $R_t$ but then skips to the next step $t + 1$. The budget at step $t + 1$ is thus $B - 1$ and the group of the best candidate seen so far remains unchanged. Given that the single threshold algorithm is memory-less, its state at time $t + 1$ is fully determined by $B - 1$, $g_t^*$ and the

number of candidates seen in each group so far, which is controlled by $\mathcal{C}_N$. We deduce that

$$
\begin{aligned}
\mathbf{P}(\mathcal{A}_T^B &\text{ succeeds}, R_t \neq 1, \rho_1 = t, g_t = \ell, g_{T-1}^* = k \mid \mathcal{C}_N) \\
&= \mathbf{P}(\mathcal{A}_T^B \text{ succeeds} \mid R_t \neq 1, \rho_1 = t, g_t = \ell, g_{T-1}^* = k, \mathcal{C}_N) \qquad (5) \\
&\quad \times \mathbf{P}(R_t \neq 1, \rho_1 = t, g_t = \ell, g_{T-1}^* = k \mid \mathcal{C}_N) \\
&= \mathbf{P}(\mathcal{A}_{t+1}^{B-1} \text{ succeeds} \mid g_t^* = k, \mathcal{C}_N)\mathbf{P}(R_t \neq 1, \rho_1 = t, g_t = \ell, g_{T-1}^* = k \mid \mathcal{C}_N), \quad (6)
\end{aligned}
$$

where $\mathbf{P}(\mathcal{A}_{N+1}^{B-1} \text{ succeeds}) = 0$. For $\ell = k$, the probability $\mathbf{P}(R_t \neq 1, \rho_1 = t, g_t = \ell, g_{T-1}^* = k \mid \mathcal{C}_N)$ is zero, because if $g_{T-1}^* = k$, $\rho_1 \geq t$ and $g_t = \ell$, then the best candidate up to step $T-1$ belongs to group $G^k$, and no candidate $x_s$ for $s \in \{T, \dots, t-1\}$ is better than the maximum in its group seen before step $T-1$, thus if $x_t$ belongs to $G^k$ and $r_t = 1$ then necessarily $R_t = 1$.

For $\ell \neq k$, it holds that

$$
\begin{aligned}
\mathbf{P}(R_t \neq 1, &\rho_1 = t, g_t = \ell, g_{T-1}^* = k \mid \mathcal{C}_N) \\
&= \mathbf{P}(\rho_1 = t, g_t = \ell, \max x_{1:t} \in G_{T-1}^k \mid \mathcal{C}_N) \\
&= \mathbf{P}(\max x_{1:t} \in G_{T-1}^k \mid \mathcal{C}_N)\mathbf{P}(\rho_1 = t, g_t = \ell \mid \max x_{1:t} \in G_{T-1}^k, \mathcal{C}_N) \\
&= \left(\frac{\lambda_k(T-1)}{t-1} + O(1/N^2)\right)\mathbf{P}(\rho_1 = t, g_t = \ell \mid \max x_{1:t} \in G_{T-1}^k, \mathcal{C}_N) \\
&= \left(\frac{\lambda_k T}{t} + O(1/N^2)\right)\mathbf{P}(g_t = \ell \mid \mathcal{C}_N)\mathbf{P}(\max G_{T:t-1}^\ell < \max G_{T-1}^\ell < x_t \\
&\qquad \text{and } \forall m \in [K] \setminus \{k, \ell\} : \max G_{T:t-1}^m < \max G_{T-1}^m \mid \mathcal{C}_N) \\
&= \left(\frac{\lambda_k T}{t} + O(1/N^2)\right)(\lambda_\ell + O(1/N^2))\mathbf{E}\left[\frac{1}{|G_{t-1}^\ell|+1} \cdot \frac{|G_{T-1}^\ell|}{|G_{t-1}^\ell|} \,\Big|\, \mathcal{C}_N\right] \prod_{m \notin \{k, \ell\}} \mathbf{E}\left[\frac{|G_{t-1}^m|}{|G_{T-1}^m|} \,\Big|\, \mathcal{C}_N\right] \\
&= \left(\frac{\lambda_\ell \lambda_k T}{t} + O(1/N^2)\right)\mathbf{E}\left[\frac{1}{|G_{t-1}^\ell|+1} \cdot \frac{|G_{T-1}^\ell|}{|G_{t-1}^\ell|} \,\Big|\, \mathcal{C}_N\right] \prod_{m \notin \{k, \ell\}} \mathbf{E}\left[\frac{|G_{t-1}^m|}{|G_{T-1}^m|} \,\Big|\, \mathcal{C}_N\right] \\
&= \left(\frac{\lambda_\ell \lambda_k T}{t} + O(1/N^2)\right)\left(\frac{T}{\lambda_\ell t^2} + O\left(\sqrt{\frac{\log N}{N^3}}\right)\right) \prod_{m \notin \{k, \ell\}}\left(\frac{T}{t} + O\left(\sqrt{\frac{\log N}{N}}\right)\right) \\
&= \left(\frac{\lambda_\ell \lambda_k T}{t} + O(1/N^2)\right)\left(\frac{T}{\lambda_\ell t^2} + O\left(\sqrt{\frac{\log N}{N^3}}\right)\right)\left(\frac{T^{K-2}}{t^{K-2}} + O\left(\sqrt{\frac{\log N}{N}}\right)\right) \\
&= \frac{\lambda_k T^K}{t^{K+1}} + O\left(\sqrt{\frac{\log N}{N^3}}\right),
\end{aligned}
$$

where we used in the last equations the event $\mathcal{C}_N$ and the assumption $T = \Theta(N)$. Therefore, substituting into (6) gives for all $\ell, k \in [K]$ and $t \in \{T, \dots, N\}$ that

$$
\begin{aligned}
\mathbf{P}(\mathcal{A}_T^B &\text{ succeeds}, R_t \neq 1, \rho_1 = t, g_t = \ell, g_{T-1}^* = k \mid \mathcal{C}_N) \\
&= \left(\frac{\lambda_k T^K}{t^{K+1}} + O\left(\sqrt{\frac{\log N}{N^3}}\right)\right)\mathbb{1}(k \neq \ell)\,\mathbf{P}(\mathcal{A}_{t+1}^{B-1} \text{ succeeds} \mid g_t^* = k, \mathcal{C}_N) \\
&= \left(\frac{\lambda_k T^K}{t^{K+1}} + O\left(\sqrt{\frac{\log N}{N^3}}\right)\right)\mathbb{1}(k \neq \ell)\frac{\mathbf{P}(\mathcal{A}_{t+1}^{B-1} \text{ succeeds}, g_t^* = k \mid \mathcal{C}_N)}{\mathbf{P}(g_t^* = k \mid \mathcal{C}_N)} \\
&= \left(\frac{\lambda_k T^K}{t^{K+1}} + O\left(\sqrt{\frac{\log N}{N^3}}\right)\right)\mathbb{1}(k \neq \ell)\frac{\mathbf{P}(\mathcal{A}_{t+1}^{B-1} \text{ succeeds}, g_t^* = k \mid \mathcal{C}_N)}{\lambda_k + O(1/N^2)} \\
&= \left(\frac{T^K}{t^{K+1}} + O\left(\sqrt{\frac{\log N}{N^3}}\right)\right)\mathbb{1}(k \neq \ell)\,\mathbf{P}(\mathcal{A}_{t+1}^{B-1} \text{ succeeds}, g_t^* = k \mid \mathcal{C}_N). \quad (7)
\end{aligned}
$$

Finally, substituting (4) and (7) into (3), and recalling that all the previous $O$ terms are independent of $t$, gives that

$$\mathbf{P}(\mathcal{A}_T^B \text{ succeeds} \mid \mathcal{C}_N) = \sum_{t=T}^{N}\sum_{k=1}^{K}\sum_{\ell=1}^{K}\left(\frac{\lambda_\ell\lambda_k}{N}(T/t)^K + O\left(\sqrt{\tfrac{\log N}{N^3}}\right)\right)$$

$$+ \sum_{t=T}^{N}\sum_{k=1}^{K}\sum_{\ell\neq k}\left(\frac{T^K}{t^{K+1}} + O\left(\sqrt{\tfrac{\log N}{N^3}}\right)\right)\mathbf{P}(\mathcal{A}_{t+1}^{B-1} \text{ succeeds}, g_t^* = k \mid \mathcal{C}_N)$$

$$= \left(1 + O\left(\sqrt{\tfrac{\log N}{N}}\right)\right)\left(\sum_{t=T}^{N}\sum_{k=1}^{K}\sum_{\ell=1}^{K}\frac{\lambda_\ell\lambda_k}{N}(T/t)^K\right.$$

$$+ \left.\sum_{t=T}^{N}\sum_{k=1}^{K}\sum_{\ell\neq k}\frac{T^K}{t^{K+1}}\mathbf{P}(\mathcal{A}_{t+1}^{B-1} \text{ succeeds}, g_t^* = k \mid \mathcal{C}_N)\right)$$

$$= \left(1 + O\left(\sqrt{\tfrac{\log N}{N}}\right)\right)\left(\frac{T^K}{N}\sum_{t=T}^{N}\frac{1}{t^K} + (K-1)\sum_{t=T}^{N}\frac{T^K}{t^{K+1}}\mathbf{P}(\mathcal{A}_{t+1}^{B-1} \text{ succeeds} \mid \mathcal{C}_N)\right).$$

Using Riemann sum properties, we obtain

$$\frac{T^K}{N}\sum_{t=T}^{N}\frac{1}{t^K} = \frac{(T/N)^K}{N}\sum_{t=T}^{N}\frac{1}{(t/N)^K} = \alpha^K\int_\alpha^1\frac{du}{u^K} + O(1/N) = \frac{\alpha - \alpha^K}{K-1} + O(1/N),$$

and by Lemma A.3 we have for all $t \in \{T, \ldots, N\}$ and $b \geq 0$ that

$$\mathbf{P}(\mathcal{A}_t^b \text{ succeeds} \mid \mathcal{C}_N) = \mathbf{P}(\mathcal{A}_t^b \text{ succeeds}) + O(1/N^2),$$

with the $O(1/N^2)$ independent of $t$. Observing that $\sum_{t=T}^{N}\frac{T^K}{t^{K+1}} = \frac{1-\alpha^K}{K} + o(1) = O(1)$, it follows that

$$\mathbf{P}(\mathcal{A}_T^B \text{ succeeds}) = \left(1 + O\left(\sqrt{\tfrac{\log N}{N}}\right)\right)\left(\frac{\alpha - \alpha^K}{K-1} + (K-1)\sum_{t=T}^{N}\frac{T^K}{t^{K+1}}\mathbf{P}(\mathcal{A}_{t+1}^{B-1} \text{ succeeds}) + O(\tfrac{1}{N})\right)$$

$$= \frac{\alpha - \alpha^K}{K-1} + (K-1)\sum_{t=T}^{N}\frac{T^K}{t^{K+1}}\mathbf{P}(\mathcal{A}_{t+1}^{B-1} \text{ succeeds}) + O\left(\sqrt{\tfrac{\log N}{N}}\right).$$

This concludes the proof for $B \geq 1$.

For $B = 0$, $\mathbf{P}(\mathcal{A}_{t+1}^0 \text{ succeeds} \mid \mathcal{C}_N)$ can be decomposed as in (3). However, the terms of the second sum are all zero, because if $\rho_1 = t$ then the algorithm stops at $t$, but since $R_t \neq 1$, the selected candidate is not the best one, and thus the succeeding probability is 0. All the computations regarding the first sum stay the same, and we obtain

$$\mathbf{P}(\mathcal{A}_T^0 \text{ succeeds}) = \frac{\alpha - \alpha^K}{K-1} + O\left(\sqrt{\tfrac{\log N}{N}}\right).$$

$\square$

## B.2 Proof of Theorem 3.2

*Proof.* Let $\alpha \in (0,1]$ a constant. For all $w \in [\alpha, 1]$ and $B \geq 0$, we denote by $\varphi^B(w)$ the limit $\lim_{N\to\infty}\mathbf{P}(\mathcal{A}_t^B \text{ succeeds})$ for $t = \lfloor wN \rfloor$. We will prove by induction over $B$ that this limit exists for all $w \in [\alpha, 1]$, is equal to the expression stated in the theorem, with $u$ instead of $\alpha$, and satisfies $\mathbf{P}(\mathcal{A}_t^B \text{ succeeds}) = \varphi^B(w) + O\left(\sqrt{\tfrac{\log N}{N}}\right)$, with the $O$ term only depending on $\alpha$ and the other constants of the problem. In particular, the $O$ term is independent of $t$. For $B = 0$, Lemma 3.1 gives immediately for any $w \in [\alpha, 1]$ and $t = \lfloor wN \rfloor$ that

$$\mathbf{P}(\mathcal{A}_t^0 \text{ succeeds}) = \frac{w - w^K}{K-1} + O\left(\sqrt{\tfrac{\log N}{N}}\right) = \frac{w^K}{K-1}\left(\frac{1}{w^{K-1}} - 1\right) + O\left(\sqrt{\tfrac{\log N}{N}}\right).$$

The $O$ term depends on $t$, but using the inequalities $\alpha + o(1) \leq t/N \leq 1$, it can be made only dependent on $\alpha$. Let $B \geq 1$ and assume the result is true for $B - 1$. Lemma 3.1 and the induction hypothesis give for all $w \in [\alpha, 1]$ and $t = \lfloor wN \rfloor$, that

$$
\mathbf{P}(\mathcal{A}_t^B \text{ succeeds}) = \frac{w - w^K}{K - 1} + (K - 1) \sum_{s=t}^{N} \frac{t^K}{s^{K+1}} \mathbf{P}(\mathcal{A}_{s+1}^{B-1} \text{ succeeds}) + O\left(\sqrt{\frac{\log N}{N}}\right)
$$

$$
= \frac{w - w^K}{K - 1} + (K - 1) \sum_{s=t}^{N} \frac{t^K}{s^{K+1}} \left( \varphi^{B-1}\left(\tfrac{s+1}{N}\right) + O\left(\sqrt{\tfrac{\log N}{N}}\right) \right) + O\left(\sqrt{\tfrac{\log N}{N}}\right)
$$

$$
= \frac{w - w^K}{K - 1} + (K - 1) \frac{(t/N)^K}{N} \sum_{s=t}^{N} \frac{\varphi^{B-1}\left(\frac{s+1}{N}\right)}{(s/N)^{K+1}} + O\left(\sqrt{\tfrac{\log N}{N}}\right),
$$

where we used that the $O$ term in the induction hypothesis is independent of $s$ and that

$$
\sum_{s=t}^{N} \frac{t^K}{s^{K+1}} \varphi^{B-1}\left(\tfrac{s+1}{N}\right) \leq \sum_{s=T}^{N} \frac{N^K}{s^{K+1}} \leq \frac{1}{N} \sum_{s=T}^{N} \frac{1}{(s/N)^{K+1}} = O(1).
$$

Finally, $t/N = w + O(1/N)$, and $\varphi^{B-1}$ is, by the induction hypothesis, a continuously differentiable function on $[\alpha, 1]$, therefore, it holds by convergence properties of Riemann sums that

$$
\mathbf{P}(\mathcal{A}_t^B \text{ succeeds}) = \frac{w - w^K}{K - 1} + (K - 1)(w^K + O(\tfrac{1}{N})) \left( \int_w^1 \frac{\varphi^{B-1}(u)}{u^{K+1}} du + O(\tfrac{1}{N}) \right) + O\left(\sqrt{\tfrac{\log N}{N}}\right)
$$

$$
= \frac{w - w^K}{K - 1} + (K - 1)w^K \int_w^1 \frac{\varphi^{B-1}(u)}{u^{K+1}} du + O\left(\sqrt{\tfrac{\log N}{N}}\right),
$$

where the $O$ term depends on $t$ and the constant parameters. Using that $T = \lfloor \alpha N \rfloor \leq t \leq N$, the $O$ can be made dependent only on $\alpha$ and the other constant parameters. The limit $\varphi^B(w) = \lim_{N \to \infty} \mathbf{P}(\mathcal{A}_{\lfloor wN \rfloor}^B \text{ succeeds})$ therefore exists, and is equal to

$$
\varphi^B(w) = \frac{w - w^K}{K - 1} + (K - 1)w^K \int_w^1 \frac{\varphi^{B-1}(u)}{u^{K+1}} du.
$$

The induction hypothesis gives for all $u \in [\alpha, 1]$ that

$$
\varphi^{B-1}(u) = \frac{u^K}{K - 1} \sum_{b=0}^{B-1} \left( \frac{1}{u^{K-1}} - \sum_{\ell=0}^{b} \frac{\log(1/u^{K-1})^\ell}{\ell!} \right),
$$

hence

$$
(K-1)w^K \int_w^1 \frac{\varphi^{B-1}(u)}{u^{K+1}} du = \int_w^1 \sum_{b=0}^{B-1} \left( \frac{1}{u^K} - \sum_{\ell=0}^{b} \frac{\log(1/u^{K-1})^\ell}{\ell! u} \right) du
$$

$$
= Bw^K \int_w^1 \frac{du}{u^K} - w^K \sum_{b=0}^{B-1} \sum_{\ell=0}^{b} \frac{1}{\ell!} \int_w^1 \frac{\log(1/u^{K-1})^\ell}{u} du
$$

$$
= Bw^K \left[ \frac{-1}{(K-1)u^{K-1}} \right]_w^1 - w^K \sum_{b=0}^{B-1} \sum_{\ell=0}^{b} \frac{1}{\ell!} \left[ -\frac{\log(1/u^{K-1})^{\ell+1}}{(K-1)(\ell+1)} \right]_w^1
$$

$$
= \frac{Bw^K}{K-1} \left( \frac{1}{w^{K-1}} - 1 \right) - w^K \sum_{b=0}^{B-1} \sum_{\ell=0}^{b} \frac{\log(1/w^{K-1})^{\ell+1}}{(K-1)(\ell+1)!}
$$

$$
= \frac{Bw^K}{K-1} \left( \frac{1}{w^{K-1}} - 1 \right) - w^K \sum_{b=1}^{B} \sum_{\ell=1}^{b} \frac{\log(1/w^{K-1})^\ell}{(K-1)\ell!}
$$

$$
= \frac{w^K}{K-1} \sum_{b=1}^{B} \left( \frac{1}{w^{K-1}} - 1 - \sum_{\ell=1}^{b} \frac{\log(1/w^{K-1})^\ell}{\ell!} \right)
$$

$$
= \frac{w^K}{K-1} \sum_{b=1}^{B} \left( \frac{1}{w^{K-1}} - \sum_{\ell=0}^{b} \frac{\log(1/w^{K-1})^\ell}{\ell!} \right),
$$

and it follows that

$$\varphi^B(w) = \frac{w^K}{K-1}\left(\frac{1}{w^{K-1}} - 1 + \sum_{b=1}^{B}\left(\frac{1}{w^{K-1}} - \sum_{\ell=0}^{b}\frac{\log(1/w^{K-1})^\ell}{\ell!}\right)\right)$$

$$= \frac{w^K}{K-1}\sum_{b=0}^{B}\left(\frac{1}{w^{K-1}} - \sum_{\ell=0}^{b}\frac{\log(1/w^{K-1})^\ell}{\ell!}\right).$$

In particular, this identity is true for $w = \alpha$, which gives the wanted result.

**Infinite budget**   Taking the limit for $B \to \infty$, we obtain that

$$\lim_{B\to\infty}\lim_{N\to\infty}\mathbf{P}(\mathcal{A}^B_{\lfloor \alpha N\rfloor}\text{ succeeds}) = \lim_{B\to\infty}\varphi^B(w)$$

$$= \frac{\alpha^K}{K-1}\sum_{b=0}^{\infty}\left(\frac{1}{\alpha^{K-1}} - \sum_{\ell=0}^{b}\frac{\log(1/\alpha^{K-1})^\ell}{\ell!}\right)$$

$$= \frac{\alpha^K}{K-1}\sum_{b=0}^{\infty}\sum_{\ell=b+1}^{\infty}\frac{\log(1/\alpha^{K-1})^\ell}{\ell!}$$

$$= \frac{\alpha^K}{K-1}\sum_{\ell=1}^{\infty}\sum_{b=0}^{\ell-1}\frac{\log(1/\alpha^{K-1})^\ell}{\ell!}$$

$$= \frac{\alpha^K}{K-1}\sum_{\ell=1}^{\infty}\frac{\log(1/\alpha^{K-1})^\ell}{(\ell-1)!}$$

$$= \frac{\alpha^K \log(1/\alpha^{K-1})}{K-1}\sum_{\ell=0}^{\infty}\frac{\log(1/\alpha^{K-1})^\ell}{\ell!}$$

$$= \alpha^K \log(1/\alpha)\cdot\frac{1}{\alpha^{K-1}}$$

$$= \alpha\log(1/\alpha).$$

$\square$

### B.3   Proof of Corollary 3.2.1

*Proof.* Let $\alpha \in (0,1)$ and $T = \lfloor \alpha N\rfloor$. Lemma A.4 gives for all $x > 0$ that

$$\sum_{b=0}^{B}\left(e^x - \sum_{\ell=0}^{b}\frac{x^\ell}{\ell!}\right) \geq xe^x\left(1 - \frac{x^{B+1}}{(B+1)!}\right),$$

in particular, we obtain for $x = \log(1/\alpha^{K-1})$ that

$$\lim_{N\to\infty}\mathbf{P}(\mathcal{A}^B_T\text{ succeeds}) = \frac{\alpha^K}{K-1}\sum_{b=0}^{\infty}\left(\frac{1}{\alpha^{K-1}} - \sum_{\ell=0}^{b}\frac{\log(1/\alpha^{K-1})^\ell}{\ell!}\right)$$

$$\geq \frac{\alpha^K}{K-1}\cdot\frac{\log(1/\alpha^{K-1})}{\alpha^{K-1}}\left(1 - \frac{\log(1/\alpha^{K-1})^{B+1}}{(B+1)!}\right)$$

$$= \alpha\log(1/\alpha)\left(1 - \frac{(K-1)^{B+1}\log(1/\alpha)^{B+1}}{(B+1)!}\right).$$

Taking a threshold $T = \lfloor N/e\rfloor$ gives

$$\lim_{N\to\infty}\mathbf{P}(\mathcal{A}^B_{\lfloor N/e\rfloor}\text{ succeeds}) \geq \frac{1}{e}\left(1 - \frac{(K-1)^{B+1}}{(B+1)!}\right).$$

To achieve an asymptotic success probability of at least $\frac{1-\varepsilon}{e}$ for some $\varepsilon > 0$, using the inequality $m! \geq e\left(\frac{m}{e}\right)^m$, it suffices that $K$ and $B$ satisfy $\frac{(K-1)^{B+1}}{e\left(\frac{B+1}{e}\right)^{B+1}} \leq \varepsilon$, which is equivalent to

$$K \leq 1 + \frac{B+1}{e}(e\varepsilon)^{\frac{1}{B+1}}.$$

$\square$

# C Static Double-threshold algorithm for two groups

## C.1 Recursion lemma

We first prove a recursion satisfied by $\mathcal{U}_{N,t,k}^B$ (2), which we use to prove the subsequent results in this section.

**Lemma C.1.** *For all $B \geq 0$, $t \in \{\lfloor \alpha N \rfloor, \dots, \lfloor \beta N \rfloor - 1\}$, and $k \in \{1, 2\}$, $\mathcal{U}_{N,t,k}^B$ satisfies*

$$
\mathcal{U}_{N,t,k}^B = \frac{\lambda}{N} \sum_{s=t}^{\lfloor \beta N \rfloor - 1} \mathbf{P}(\rho_1 \geq s, g_{t-1}^* = k \mid \mathcal{C}_N)
$$
$$
+ \frac{\mathbb{1}(B > 0)}{1 - \lambda} \sum_{s=t}^{\lfloor \beta N \rfloor - 1} \mathbf{P}(g_{t-1}^* = k, \rho_1 = s, R_s \neq 1 \mid \mathcal{C}_N) \mathcal{U}_{N,s+1,2}^{B-1}
$$
$$
+ \mathbf{P}(\mathcal{A}_t^B(\alpha, \beta) \text{ succeeds}, g_{t-1}^* = k, \rho_1 \geq \beta N \mid \mathcal{C}_N) + O\left(\frac{1}{N}\right) .
$$

Assuming that $\rho_1 = s \in \{t, \dots, \lfloor \beta N \rfloor - 1\}$, the first sum corresponds to the success probability if $R_s = 1$ and the algorithm selects the candidate $x_s$. The terms of the second sum represent the success probability after using a comparison at step $s$ but observing $R_t \neq 1$, resulting in the rejection of the candidate. Therefore, the available budget at step $s + 1$ is $B - 1$, and necessarily $g_s^* = 2$, because a comparison at step $s$ can only occur if $g_s = 1$ by definition of the algorithm. Hence, only the term $\mathcal{U}_{N,s+1,2}^{B-1}$ appears in the recursion, not $\mathcal{U}_{N,s+1,1}^{B-1}$. Finally, the last term represents the probability of success if no comparison has been made before step $\lfloor \beta N \rfloor$

*Proof.* Let $B \geq 0$. For all $t \in \{\lfloor \alpha N \rfloor, \dots, \lfloor \beta N \rfloor - 1)$ and $k \in \{1, 2\}$, it holds that

$$
\mathcal{U}_{N,t,k}^B = \mathbf{P}(\mathcal{A}_t^B(\alpha, \beta) \text{ succeeds}, g_{t-1}^* = k \mid \mathcal{C}_N)
$$
$$
= \sum_{s=t}^{\lfloor \beta N \rfloor - 1} \mathbf{P}(\mathcal{A}_t^B(\alpha, \beta) \text{ succeeds}, g_{t-1}^* = k, \rho_1 = s \mid \mathcal{C}_N)
$$
$$
+ \mathbf{P}(\mathcal{A}_t^B(\alpha, \beta) \text{ succeeds}, g_{t-1}^* = k, \rho_1 \geq \beta N \mid \mathcal{C}_N)
$$
$$
= \sum_{s=t}^{\lfloor \beta N \rfloor - 1} \mathbf{P}(\mathcal{A}_t^B(\alpha, \beta) \text{ succeeds}, g_{t-1}^* = k, \rho_1 = s, R_s = 1 \mid \mathcal{C}_N) \tag{8}
$$
$$
+ \sum_{s=t}^{\lfloor \beta N \rfloor - 1} \mathbf{P}(\mathcal{A}_t^B(\alpha, \beta) \text{ succeeds}, g_{t-1}^* = k, \rho_1 = s, R_s \neq 1 \mid \mathcal{C}_N) \tag{9}
$$
$$
+ \mathbf{P}(\mathcal{A}_t^B(\alpha, \beta) \text{ succeeds}, g_{t-1}^* = k, \rho_1 \geq \beta N \mid \mathcal{C}_N) . \tag{10}
$$

For all $s \in \{t, \dots, \lfloor \beta N \rfloor - 1\}$, by definition of Algorithm $\mathcal{A}_t^B(\alpha, \beta)$, if $\rho_1 = s$ and $R_s = 1$ then the candidate $x_s$ is selected, and the algorithm succeeds if only if $x_s = x_{\max}$. Moreover, the event $\{\rho_1 = s\}$ is equivalent to $\{\rho_1 \geq s, g_s = 1, r_s = 1\}$, hence the terms in (8) can be written as

$$
\mathbf{P}(\mathcal{A}_t^B(\alpha, \beta) \text{ succeeds}, g_{t-1}^* = k, \rho_1 = s, R_s = 1 \mid \mathcal{C}_N)
$$
$$
= \mathbf{P}(x_s = x_{\max}, g_{t-1}^* = k, \rho_1 = s, R_s = 1 \mid \mathcal{C}_N)
$$
$$
= \mathbf{P}(x_s = x_{\max}, g_{t-1}^* = k, \rho_1 \geq s, g_s = 1 \mid \mathcal{C}_N)
$$
$$
= \frac{\mathbf{P}(g_s = 1 \mid \mathcal{C}_N)}{N} \mathbf{P}(\rho_1 \geq s, g_{t-1}^* = k \mid \mathcal{C}_N)
$$
$$
= \left( \frac{\lambda}{N} + O\left(\frac{1}{N^3}\right) \right) \mathbf{P}(\rho_1 \geq s, g_{t-1}^* = k \mid \mathcal{C}_N) ,
$$

where used for that the event $\{x_s = x_{\max}\}$ is independent of the group memberships, thus independent of $\mathcal{C}_N$, and that it is also independent of $\{\rho_1 \geq s, g^*_{t-1} = k\}$, because a realization of the latter event is determined only by the groups and relative ranks of the candidates $\{x_1, \dots, x_{s-1}\}$. For the last equality, we used Lemma A.3.

Secondly, in the case where $\rho_1 = s$ and $R_s \neq 1$, if $B = 0$ then the algorithm selects candidate $x_s$ which is not the best overall, hence its probability of succeeding is zero. If $B \geq 1$, the algorithm makes a comparison but then rejects the candidate. Moreover, for $s \in [t, \beta N]$, if $\rho_1 = s$ then necessarily $g_s = 1$, and having $R_s \neq 1$ implies that $g^*_s = 2$. The success probability of $\mathcal{A}^B_t(\alpha, \beta)$ given that $\rho_1 = s, R_s \neq 1$ is the same as the success probability of $\mathcal{A}^{B-1}_{s+1}(\alpha, \beta)$ given that $g^*_s = 2$. Therefore, the terms of (9) satisfy

$$
\mathbf{P}(\mathcal{A}^B_t(\alpha, \beta) \text{ succeeds}, g^*_{t-1} = k, \rho_1 = s, R_s \neq 1 \mid \mathcal{C}_N)
$$
$$
= \mathbb{1}(B > 0)\, \mathbf{P}(g^*_{t-1} = k, \rho_1 = s, R_s \neq 1 \mid \mathcal{C}_N)\mathbf{P}(\mathcal{A}^B_t(\alpha, \beta) \text{ succeeds} \mid g^*_{t-1} = k, \rho_1 = s, R_s \neq 1, \mathcal{C}_N)
$$
$$
= \mathbb{1}(B > 0)\, \mathbf{P}(g^*_{t-1} = k, \rho_1 = s, R_s \neq 1 \mid \mathcal{C}_N)\mathbf{P}(\mathcal{A}^{B-1}_{s+1}(\alpha, \beta) \text{ succeeds} \mid g^*_s = 2, \mathcal{C}_N)
$$
$$
= \mathbb{1}(B > 0)\, \mathbf{P}(g^*_{t-1} = k, \rho_1 = s, R_s \neq 1 \mid \mathcal{C}_N)\frac{\mathbf{P}(\mathcal{A}^{B-1}_{s+1}(\alpha, \beta) \text{ succeeds}, g^*_s = 2 \mid \mathcal{C}_N)}{\mathbf{P}(g^*_s = 2 \mid \mathcal{C}_N)}
$$
$$
= \mathbb{1}(B > 0)\, \mathbf{P}(g^*_{t-1} = k, \rho_1 = s, R_s \neq 1 \mid \mathcal{C}_N)\frac{\mathcal{U}^{B-1}_{N,s+1,2}}{1 - \lambda + O(\frac{1}{N^2})}\,,
$$

where we used again Lemma A.3. Given that the $O$ terms are independent of $s$, We deduce that

$$
\mathcal{U}^B_{N,t,k} = \left(\frac{\lambda}{N} + O(\tfrac{1}{N^3})\right)\sum_{s=t}^{\lfloor \beta N \rfloor - 1} \mathbf{P}(\rho_1 \geq s, g^*_{t-1} = k \mid \mathcal{C}_N)
$$
$$
+ \mathbb{1}(B > 0)\left(\frac{1}{1 - \lambda} + O(\tfrac{1}{N^2})\right)\sum_{s=t}^{\lfloor \beta N \rfloor - 1} \mathbf{P}(g^*_{t-1} = k, \rho_1 = s, R_s \neq 1 \mid \mathcal{C}_N)\mathcal{U}^{B-1}_{N,s+1,2}
$$
$$
+ \mathbf{P}(\mathcal{A}^B_t(\alpha, \beta) \text{ succeeds}, g^*_{t-1} = k, \rho_1 \geq \beta N \mid \mathcal{C}_N)
$$
$$
= \frac{\lambda}{N}\sum_{s=t}^{\lfloor \beta N \rfloor - 1} \mathbf{P}(\rho_1 \geq s, g^*_{t-1} = k \mid \mathcal{C}_N)
$$
$$
+ \frac{\mathbb{1}(B > 0)}{1 - \lambda}\sum_{s=t}^{\lfloor \beta N \rfloor - 1} \mathbf{P}(g^*_{t-1} = k, \rho_1 = s, R_s \neq 1 \mid \mathcal{C}_N)\mathcal{U}^{B-1}_{N,s+1,2}
$$
$$
+ \mathbf{P}(\mathcal{A}^B_t(\alpha, \beta) \text{ succeeds}, g^*_{t-1} = k, \rho_1 \geq \beta N \mid \mathcal{C}_N) + O(\tfrac{1}{N})\,.
$$

$\square$

## C.2 Additional lemmas

In the following two lemmas, we compute the probabilities appearing in Lemma C.1 for all $s \in \{t, \dots, \lfloor \beta N \rfloor - 1\}$ and $k \in \{1, 2\}$

**Lemma C.2.** *Let $\lfloor \alpha N \rfloor \leq t \leq s < \lfloor \beta N \rfloor$, and consider a run of Algorithm $\mathcal{A}^B_t(\alpha, \beta)$, then it holds that*

$$
\mathbf{P}(\rho_1 \geq s, g^*_{t-1} = 1 \mid \mathcal{C}_N) = \frac{\lambda t}{(1 - \lambda)t + \lambda s} + O\left(\sqrt{\tfrac{\log N}{N}}\right)
$$
$$
\mathbf{P}(\rho_1 \geq s, g^*_{t-1} = 2 \mid \mathcal{C}_N) = \frac{(1 - \lambda)t^2}{((1 - \lambda)t + \lambda s)s} + O\left(\sqrt{\tfrac{\log N}{N}}\right).
$$

*Proof.* Since there are only 2 groups, the event $g^*_{t-1} = 1$ is equivalent to $\max G^2_{t-1} < \max G^1_{t-1}$. For $s \in \{t, \dots, \lfloor \beta N \rfloor\}$, Algorithm $\mathcal{A}^B_t(\alpha, \beta)$ only makes a comparison (or stops in the case of $B = 0$) at step $s$ only if $g_s = 1$ and $r_s = 1$. Therefore, $\rho_1 \geq s$ if and only if no candidate belonging

to $G^1_{t:s-1}$ surpasses the maximum value observed in $G^1_{t-1}$

$$\mathbf{P}(\rho_1 \geq s, g^*_{t-1} = 1 \mid \mathcal{C}_N) = \mathbf{P}(\max G^1_{t:s-1} < G^1_{t-1}, \max G^2_{t-1} < \max G^1_{t-1} \mid \mathcal{C}_N)$$

$$= \mathbf{P}(\max(G^1_{t:s-1} \cup G^2_{t-1}) < G^1_{t-1} \mid \mathcal{C}_N)$$

$$= \mathbf{E}\left[ \frac{|G^1_{t-1}|}{|G^1_{t-1}| + |G^1_{t:s-1}| + |G^2_{t-1}|} \,\middle|\, \mathcal{C}_N \right]$$

$$= \mathbf{E}\left[ \frac{|G^1_{t-1}|}{t - 1 + |G^1_{t:s-1}|} \,\middle|\, \mathcal{C}_N \right]$$

$$= \frac{\lambda t + O(\sqrt{N \log N})}{t + \lambda(s - t) + O(\sqrt{N \log N})}$$

$$= \frac{\lambda t}{(1 - \lambda)t + \lambda s} + O\left(\sqrt{\tfrac{\log N}{N}}\right).$$

For the case $g^*_t = 2$, we obtain

$$\mathbf{P}(\rho_1 \geq s, g^*_{t-1} = 2 \mid \mathcal{C}_N) = \mathbf{P}(\max G^1_{t:s-1} < G^1_{t-1}, \max G^1_{t-1} < \max G^2_{t-1} \mid \mathcal{C}_N)$$

$$= \max G^1_{t:s-1} < G^1_{t-1} < \max G^2_{t-1} \mid \mathcal{C}_N)$$

$$= \mathbf{E}\left[ \frac{|G^2_{t-1}|}{|G^1_{t-1}| + |G^1_{t:s-1}| + |G^2_{t-1}|} \cdot \frac{|G^1_{t-1}|}{|G^1_{t:s-1}| + |G^1_{t-1}|} \,\middle|\, \mathcal{C}_N \right]$$

$$= \mathbf{E}\left[ \frac{|G^2_{t-1}|}{t - 1 + |G^1_{t:s-1}|} \cdot \frac{|G^1_{t-1}|}{|G^1_{s-1}|} \,\middle|\, \mathcal{C}_N \right]$$

$$= \frac{(1 - \lambda)t + O(\sqrt{N \log N})}{t + \lambda(s - t) + O(\sqrt{N \log N})} \cdot \frac{\lambda t + O(\sqrt{N \log N})}{\lambda s + O(\sqrt{N \log N})}$$

$$= \frac{(1 - \lambda)t^2}{((1 - \lambda)t + \lambda s)s} + O\left(\sqrt{\tfrac{\log N}{N}}\right).$$

$\square$

**Lemma C.3.** *Let $\lfloor \alpha N \rfloor \leq t \leq s < \lfloor \beta N \rfloor$, and consider a run of Algorithm $\mathcal{A}^B_t(\alpha, \beta)$, then*

$$\mathbf{P}(\rho_1 = s, R_s \neq 1, g^*_{t-1} = 1 \mid \mathcal{C}_N) = \frac{\lambda^2(1 - \lambda)(s - t)t}{((1 - \lambda)t + \lambda s)^2 s} + O\left(\sqrt{\tfrac{\log N}{N^3}}\right)$$

$$\mathbf{P}(\rho_1 = s, R_s \neq 1, g^*_{t-1} = 2 \mid \mathcal{C}_N) = \frac{(1 - \lambda)\left((1 - \lambda)^2 t + \lambda(2 - \lambda)s\right)t^2}{((1 - \lambda)t + \lambda s)^2 s^2} + O\left(\sqrt{\tfrac{\log N}{N^3}}\right).$$

*Proof.* For Algorithm $\mathcal{A}^B_t(\alpha, \beta)$ and $s \in \{t, \ldots, \lfloor \beta N \rfloor - 1\}$, $\rho_1 = s$ if and only if $x_s$ is the first element in $G^1$ since step $t$ for which $r_s = 1$, thus

$$\rho_1 = s \iff g_s = 1 \text{ and } \max G^1_{t:s-1} < \max G^1_{t-1} < x_s.$$

Furthermore, Lemma A.3 gives that $\mathbf{P}(g_s = 1 \mid \mathcal{C}_N) = \mathbf{P}(g_s = 1) + O(1/N) = \lambda + O(1/N)$. Therefore, it holds that

$$\mathbf{P}(\rho_1 = s, R_s \neq 1, g^*_{t-1} = 1 \mid \mathcal{C}_N)$$

$$= \mathbf{P}(g_s = 1, \ \max G^1_{t:s-1} < \max G^1_{t-1} < x_s, \ x_s < \max G^2_{s-1}, \ \max G^2_{t-1} < \max G^1_{t-1} \mid \mathcal{C}_N)$$

$$= \mathbf{P}(g_s = 1 \mid \mathcal{C}_N)\mathbf{P}(\max G^1_{t:s-1} < \max G^1_{t-1} < x_s < \max G^2_{t:s-1}, \max G^2_{t-1} < \max G^1_{t-1} \mid \mathcal{C}_N)$$

$$= \mathbf{P}(g_s = 1 \mid \mathcal{C}_N)\mathbf{P}(\max(G^1_{t:s-1} \cup G^2_{t-1}) < \max G^1_{t-1} < x_s < \max G^2_{t:s-1} \mid \mathcal{C}_N)$$

$$= (\lambda + O(\tfrac{1}{N}))\mathbf{E}\left[ \frac{|G^2_{t:s-1}|}{|G^1_{t:s-1}| + |G^2_{t-1}| + |G^1_{t-1}| + 1 + |G^2_{t:s-1}|} \cdot \frac{1}{|G^1_{t:s-1}| + |G^2_{t-1}| + |G^1_{t-1}| + 1} \cdot \frac{|G^1_{t-1}|}{|G^1_{t:s-1}| + |G^2_{t-1}| + |G^1_{t-1}|} \,\middle|\, \mathcal{C}_N \right]$$

$$= (\lambda + O(\tfrac{1}{N}))\mathbf{E}\left[ \frac{|G^2_{t:s-1}|}{s} \cdot \frac{1}{t + |G^1_{t:s-1}|} \cdot \frac{|G^1_{t-1}|}{t - 1 + |G^1_{t:s-1}|} \,\middle|\, \mathcal{C}_N \right]$$

$$= (\lambda + O(\tfrac{1}{N}))\frac{(1 - \lambda)(s - t) + O(\sqrt{N \log N})}{s(t + \lambda(s - t) + O(\sqrt{N \log N}))} \cdot \frac{\lambda t + O(\sqrt{N \log N})}{t + \lambda(s - t) + O(\sqrt{N \log N})}$$

$$= \frac{\lambda^2(1 - \lambda)(s - t)t}{((1 - \lambda)t + \lambda s)^2 s} + O\left(\sqrt{\tfrac{\log N}{N^3}}\right).$$

On the other hand, in the case where $g^*_{t-1} = 2$, we obtain

$$\mathbf{P}(\rho_1 = s, R_s \neq 1, g^*_{t-1} = 2 \mid \mathcal{C}_N)$$
$$= \mathbf{P}(g_s = 1 \,,\, \max G^1_{t:s-1} < \max G^1_{t-1} < x_s \,,\, x_s < \max G^2_{s-1} \,,\, \max G^1_{t-1} < \max G^2_{t-1} \mid \mathcal{C}_N)$$
$$= \mathbf{P}(g_s = 1 \mid C_N)\mathbf{P}(\max G^1_{t:s-1} < \max G^1_{t-1} < x_s < \max G^2_{s-1} \,,\, \max G^1_{t-1} < \max G^2_{t-1} \mid \mathcal{C}_N)$$
$$= \mathbf{P}(g_s = 1 \mid C_N)\mathbf{P}(a < b < x_s < \max(c,d) \,,\, b < c \mid \mathcal{C}_N) \,,$$

where $a = \max G^1_{t:s-1}$, $b = \max G^1_{t-1}$, $c = \max G^2_{t-1}$ and $d = \max G^2_{t:s-1}$. Let us denote by $\mathcal{E}$ the event $\{a < b < x_s < \max(c,d)\} \cap \{b < c\}$. It holds that

$$\mathcal{E} \cap \{c < d\} = \{a < b < x_s < \max(c,d)\} \cap \{b < c\} \cap \{c < d\}$$
$$= \{a < b < c < x_s < d\} \cup \{a < b < x_s < c < d\}$$
$$= \{a < b < c < x_s < d\} \cup \left(\{a < b < x_s < c\} \cap \{c < d\}\right) \,,$$
$$\mathcal{E} \cap \{d < c\} = \{a < b < x_s < \max(c,d)\} \cap \{b < c\} \cap \{d < c\}$$
$$= \{a < b < x_s < c\} \cap \{d < c\} \,,$$

which yields

$$\mathcal{E} = \left(\mathcal{E} \cap \{c < d\}\right)\left(\mathcal{E} \cap \{d < c\}\right)$$
$$= \{a < b < c < x_s < d\} \cup \left(\{a < b < x_s < c\} \cap \{c < d\}\right) \cup \left(\{a < b < x_s < c\} \cap \{d < c\}\right)$$
$$= \{a < b < c < x_s < d\} \cup \{a < b < x_s < c\} \,.$$

The two events above are disjoint, and we have

$$\mathbf{P}(a < b < c < x_s < d \mid \mathcal{C}_N)$$
$$= \mathbf{P}(\max G^1_{t:s-1} < \max G^1_{t-1} < \max G^2_{t-1} < x_s < G^2_{t:s-1} \mid \mathcal{C}_N)$$
$$= \mathbf{E}\left[\frac{|G^2_{t:s-1}|}{|G^1_{t:s-1}|+|G^1_{t-1}|+|G^2_{t-1}|+1+|G^2_{t:s-1}|} \cdot \frac{1}{|G^1_{t:s-1}|+|G^1_{t-1}|+|G^2_{t-1}|+1} \cdot \frac{|G^2_{t-1}|}{|G^1_{t:s-1}|+|G^1_{t-1}|+|G^2_{t-1}|} \cdot \frac{|G^1_{t-1}|}{|G^1_{t:s-1}|+|G^1_{t-1}|} \cdot \mid \mathcal{C}_N\right]$$
$$= \mathbf{E}\left[\frac{|G^2_{t:s-1}|}{s} \cdot \frac{1}{t+|G^1_{t:s-1}|} \cdot \frac{|G^2_{t-1}|}{t-1+|G^1_{t:s-1}|} \cdot \frac{|G^1_{t-1}|}{|G^1_{s-1}|} \mid \mathcal{C}_N\right]$$
$$= \frac{(1-\lambda)(s-t)+O(\sqrt{N \log N})}{s(t+\lambda(s-t)+O(\sqrt{N \log N}))} \cdot \frac{(1-\lambda)t+O(\sqrt{N \log N})}{t+\lambda(s-t)+O(\sqrt{N \log N})} \cdot \frac{\lambda t+O(\sqrt{N \log N})}{\lambda s+O(\sqrt{N \log N})}$$
$$= \frac{(1-\lambda)^2(s-t)t^2}{((1-\lambda)t+\lambda s)^2 s^2} + O\left(\sqrt{\tfrac{\log N}{N^3}}\right) \,.$$

The probability of the second event is

$$\mathbf{P}(a < b < x_s < c \mid \mathcal{C}_N) = \mathbf{P}(\max G^1_{t:s-1} < \max G^1_{t-1} < x_s < \max G^2_{t-1} \mid \mathcal{C}_N)$$
$$= \mathbf{E}\left[\frac{|G^2_{t-1}|}{|G^1_{t:s-1}|+|G^1_{t-1}|+1+|G^2_{t-1}|} \cdot \frac{1}{|G^1_{t:s-1}|+|G^1_{t-1}|+1} \cdot \frac{|G^1_{t-1}|}{|G^1_{t:s-1}|+|G^1_{t-1}|} \mid \mathcal{C}_N\right]$$
$$= \mathbf{E}\left[\frac{|G^2_{t-1}|}{t+|G^1_{t:s-1}|} \cdot \frac{1}{|G^1_{s-1}|+1} \cdot \frac{|G^1_{t-1}|}{|G^1_{s-1}|} \mid \mathcal{C}_N\right]$$
$$= \frac{(1-\lambda)t+O(\sqrt{N \log N})}{t+\lambda(s-t)+O(\sqrt{N \log N})} \cdot \frac{1}{\lambda s+O(\sqrt{N \log N})} \cdot \frac{\lambda t+O(\sqrt{N \log N})}{\lambda s+O(\sqrt{N \log N})}$$
$$= \frac{(1-\lambda)t^2}{\lambda((1-\lambda)t+\lambda s)s^2} + O\left(\sqrt{\tfrac{\log N}{N^3}}\right) \,.$$

Finally, Lemma A.3 shows that $\mathbf{P}(g_s = 1 \mid C_N) = \lambda + O(1/N^2)$, and we deduce

$$
\begin{aligned}
\mathbf{P}(\rho_1 = s, R_s \neq 1, g^*_{t-1} = 2 \mid C_N) &= \lambda \mathbf{P}(\mathcal{E} \mid C_N) + O(\tfrac{1}{N^2}) \\
&= \lambda \mathbf{P}(a < b < c < x_s < d \mid C_N) + \lambda \mathbf{P}(a < b < x_s < c \mid C_N) + O(\tfrac{1}{N^2}) \\
&= \frac{\lambda(1-\lambda)^2(s-t)t^2}{((1-\lambda)t + \lambda s)^2 s^2} + \frac{\lambda(1-\lambda)t^2}{\lambda((1-\lambda)t + \lambda s)s^2} + O\left(\sqrt{\tfrac{\log N}{N^3}}\right) \\
&= \frac{(1-\lambda)t^2}{((1-\lambda)t + \lambda s)^2 s^2}\left(\lambda(1-\lambda)(s-t) + (1-\lambda)t + \lambda s\right) + O\left(\sqrt{\tfrac{\log N}{N^3}}\right) \\
&= \frac{(1-\lambda)t^2}{((1-\lambda)t + \lambda s)^2 s^2}\left((1-\lambda)^2 t + \lambda(2-\lambda)s\right) + O\left(\sqrt{\tfrac{\log N}{N^3}}\right).
\end{aligned}
$$

$\square$

**Lemma C.4.** *Let $\lfloor \alpha N \rfloor \leq t < \lfloor \beta N \rfloor$, and consider a run of Algorithm $\mathcal{A}^B_t(\alpha, \beta)$, then*

$$
\begin{aligned}
&\mathbf{P}(\mathcal{A}^B_t(\alpha, \beta) \text{ succeeds}, \rho_1 \geq \beta N, g^*_{t-1} = 1 \mid C_N) \\
&\qquad = \frac{t}{\beta N}\mathcal{U}^B_{N,\lfloor \beta N \rfloor, 1} + \frac{\lambda(\beta N - t)t}{((1-\lambda)t + \lambda \beta N)\beta N}\mathcal{U}^B_{N,\lfloor \beta N \rfloor, 2} + O\left(\sqrt{\tfrac{\log N}{N}}\right), \\
&\mathbf{P}(\mathcal{A}^B_t(\alpha, \beta) \text{ succeeds}, \rho_1 \geq \beta N, g^*_{t-1} = 2 \mid C_N) \\
&\qquad = \frac{t^2}{((1-\lambda)t + \lambda \beta N)\beta N}\mathcal{U}^B_{N,\lfloor \beta N \rfloor, 2} + O\left(\sqrt{\tfrac{\log N}{N}}\right).
\end{aligned}
$$

*Proof.* Since Algorithm $\mathcal{A}^B_t(\alpha, \beta)$ is memoryless, if it does not stop before step $\lfloor \beta N \rfloor$, then its success probability is the same as that of $\mathcal{A}^B_{\lfloor \beta N \rfloor}(\alpha, \beta) = \mathcal{A}^B_0(\beta, \beta)$, which has the same threshold $\beta$ for both groups, if it is in the same state $(g^*_{\lfloor \beta N \rfloor - 1}, |G^1_{\lfloor \beta N \rfloor}|)$. In all the proof, $\rho_1$ is relative to Algorithm $\mathcal{A}^B_t(\alpha, \beta)$, not $\mathcal{A}^B_0(\beta, \beta)$. It holds that

$$
\begin{aligned}
&\mathbf{P}(\mathcal{A}^B_t(\alpha, \beta) \text{ succeeds}, \rho_1 \geq \beta N, g^*_{t-1} = 1 \mid C_N) \\
&= \sum_{\ell \in \{1,2\}} \mathbf{P}(\mathcal{A}^B_t(\alpha, \beta) \text{ succeeds}, \rho_1 \geq \beta N, g^*_{t-1} = 1, g^*_{\lfloor \beta N \rfloor - 1} = \ell \mid C_N) \\
&= \sum_{\ell \in \{1,2\}} \mathbf{P}(\rho_1 \geq \beta N, g^*_{t-1} = 1, g^*_{\lfloor \beta N \rfloor - 1} = \ell \mid C_N)) \\
&\qquad\qquad \times \mathbf{P}(\mathcal{A}^B_t(\alpha, \beta) \text{ succeeds} \mid \rho_1 \geq \beta N, g^*_{t-1} = 1, g^*_{\lfloor \beta N \rfloor - 1} = \ell, C_N) \\
&= \sum_{\ell \in \{1,2\}} \mathbf{P}(\rho_1 \geq \beta N, g^*_{t-1} = 1, g^*_{\lfloor \beta N \rfloor - 1} = \ell \mid C_N)\mathbf{P}(\mathcal{A}^B_0(\beta, \beta) \text{ succeeds} \mid g^*_{\lfloor \beta N \rfloor - 1} = \ell, C_N) \\
&= \sum_{\ell \in \{1,2\}} \mathbf{P}(\rho_1 \geq \beta N, g^*_{t-1} = 1, g^*_{\lfloor \beta N \rfloor - 1} = \ell \mid C_N)\frac{\mathbf{P}(\mathcal{A}^B_0(\beta, \beta) \text{ succeeds}, g^*_{\lfloor \beta N \rfloor - 1} = \ell \mid C_N)}{\mathbf{P}(g^*_{\lfloor \beta N \rfloor - 1} = \ell \mid C_N)} \\
&= \sum_{\ell \in \{1,2\}} \mathbf{P}(\rho_1 \geq \beta N, g^*_{t-1} = 1, g^*_{\lfloor \beta N \rfloor - 1} = \ell \mid C_N)\frac{\mathcal{U}^B_{N,\lfloor \beta N \rfloor, \ell}}{\mathbf{P}(g^*_{\lfloor \beta N \rfloor - 1} = \ell) + O(\tfrac{1}{N^2})},
\end{aligned}
$$

where we used Lemma A.3 and the definition (2) of $\mathcal{U}_{N,s,\ell}^B$. Let us now compute the probability of the event $\{\rho_1 \geq \beta N, g_{t-1}^* = 1, g_{\lfloor \beta N \rfloor - 1}^* = \ell\}$ conditional to $\mathcal{C}_N$. For $\ell = 1$, we have

$$
\begin{aligned}
&\mathbf{P}(\rho_1 \geq \beta N, g_{t-1}^* = 1, g_{\lfloor \beta N \rfloor - 1}^* = 1 \mid \mathcal{C}_N) \\
&= \mathbf{P}(\max G_{t:\lfloor \beta N \rfloor - 1}^1 < \max G_{t-1}^1 \,,\; \max G_{t-1}^2 < \max G_{t-1}^1 \,,\; \max G_{\lfloor \beta N \rfloor - 1}^2 < \max G_{\lfloor \beta N \rfloor - 1}^1 \mid \mathcal{C}_N) \\
&= \mathbf{P}(\max x_{1:\lfloor \beta N \rfloor - 1} \in G_{t-1}^1 \mid \mathcal{C}_N) \\
&= \mathbf{E}\left[\frac{|G_{t-1}^1|}{\lfloor \beta N \rfloor - 1}\Big| \mathcal{C}_N\right] \\
&= \frac{\lambda t + O(\sqrt{N \log N})}{\beta N + O(1)} = \frac{\lambda t}{\beta N} + O\left(\sqrt{\tfrac{\log N}{N}}\right).
\end{aligned}
$$

For $\ell = 2$, we first compute the following

$$
\begin{aligned}
\mathbf{P}(\rho_1 \geq \beta N, g_{t-1}^* = 1 \mid \mathcal{C}_N) &= \mathbf{P}(\max G_{t:\lfloor \beta N \rfloor - 1}^1 < \max G_{t-1}^1 \,,\; \max G_{t-1}^2 < \max G_{t-1}^1 \mid \mathcal{C}_N) \\
&= \mathbf{P}(\max(G_{t:\lfloor \beta N \rfloor - 1}^1 \cup G_{t-1}^2) < \max G_{t-1}^1 \mid \mathcal{C}_N) \\
&= \mathbf{E}\left[\frac{|G_{t-1}^1|}{|G_{t:\lfloor \beta N \rfloor - 1}^1| + |G_{t-1}^2| + |G_{t-1}^1|}\,\Big|\, \mathcal{C}_N\right] \\
&= \frac{\lambda t + O(\sqrt{N \log N})}{t + \lambda(\beta N - t) + O(\sqrt{N \log N})} \\
&= \frac{\lambda t}{(1 - \lambda)t + \lambda \beta N} + O\left(\sqrt{\tfrac{\log N}{N}}\right),
\end{aligned}
$$

and it follows that

$$
\begin{aligned}
&\mathbf{P}(\rho_1 \geq \beta N, g_{t-1}^* = 1, g_{\lfloor \beta N \rfloor - 1}^* = 2 \mid \mathcal{C}_N) \\
&= \mathbf{P}(\rho_1 \geq \beta N, g_{t-1}^* = 1 \mid \mathcal{C}_N) - \mathbf{P}(\rho_1 \geq \beta N, g_{t-1}^* = 1, g_{\lfloor \beta N \rfloor - 1}^* = 1 \mid \mathcal{C}_N) \\
&= \frac{\lambda t}{(1 - \lambda)t + \lambda \beta N} - \frac{\lambda t}{\beta N} + O\left(\sqrt{\tfrac{\log N}{N}}\right) \\
&= \frac{\lambda(1 - \lambda)(\beta N - t)t}{((1 - \lambda)t + \lambda \beta N)\beta N} + O\left(\sqrt{\tfrac{\log N}{N}}\right).
\end{aligned}
$$

All in all, we deduce that

$$
\begin{aligned}
&\mathbf{P}(\mathcal{A}_t^B(\alpha, \beta) \text{ succeeds}, \rho_1 \geq \beta N, g_{t-1}^* = 1 \mid \mathcal{C}_N) \\
&= \left(\frac{\lambda t}{\beta N} + O\left(\sqrt{\tfrac{\log N}{N}}\right)\right) \frac{\mathcal{U}_{N,\lfloor \beta N \rfloor, 1}^B}{\lambda + O(\tfrac{1}{N^2})} + \left(\frac{\lambda(1 - \lambda)(\beta N - t)t}{((1 - \lambda)t + \lambda \beta N)\beta N} + O\left(\sqrt{\tfrac{\log N}{N}}\right)\right) \frac{\mathcal{U}_{N,\lfloor \beta N \rfloor, 2}^B}{1 - \lambda + O(\tfrac{1}{N^2})} \\
&= \left(\frac{t}{\beta N} + O\left(\sqrt{\tfrac{\log N}{N}}\right)\right) \mathcal{U}_{N,\lfloor \beta N \rfloor, 1}^B + \left(\frac{\lambda(\beta N - t)t}{((1 - \lambda)t + \lambda \beta N)\beta N} + O\left(\sqrt{\tfrac{\log N}{N}}\right)\right) \mathcal{U}_{N,\lfloor \beta N \rfloor, 2}^B \\
&= \frac{t}{\beta N} \mathcal{U}_{N,\lfloor \beta N \rfloor, 1}^B + \frac{\lambda(\beta N - t)t}{((1 - \lambda)t + \lambda \beta N)\beta N} \mathcal{U}_{N,\lfloor \beta N \rfloor, 2}^B + O\left(\sqrt{\tfrac{\log N}{N}}\right).
\end{aligned}
$$

On the other hand, if $g^*_{t-1} = 2$ and $\rho_1 \geq \beta N$, then necessarily $g^*_{\lfloor \beta N \rfloor - 1} = 2$, because no candidate in $G^1$ up to step $\lfloor \beta N \rfloor - 1$ surpasses $\max G^1_{t-1}$, which is less than $\max G^2_{t-1}$. Therefore

$$\mathbf{P}(\mathcal{A}^B_t(\alpha, \beta) \text{ succeeds}, \rho_1 \geq \beta N, g^*_{t-1} = 2 \mid \mathcal{C}_N)$$

$$= \mathbf{P}(\rho_1 \geq \beta N, g^*_{t-1} = 2 \mid \mathcal{C}_N) \mathbf{P}(\mathcal{A}^B_t(\alpha, \beta) \mid \rho_1 \geq \beta N, g^*_{t-1} = 2, \mathcal{C}_N)$$

$$= \mathbf{P}(\max G^1_{t:\lfloor \beta N \rfloor - 1} < \max G^1_{t-1} < \max G^2_{t-1} \mid \mathcal{C}_N) \mathbf{P}(\mathcal{A}^B_0(\beta, \beta) \mid g^*_{\lfloor \beta N \rfloor - 1} = 2, \mathcal{C}_N)$$

$$= \mathbf{E}\left[ \frac{|G^2_{t-1}|}{t - 1 + |G^1_{t:\lfloor \beta N \rfloor - 1}|} \cdot \frac{|G^1_{t-1}|}{|G^1_{\lfloor \beta N \rfloor - 1}|} \;\middle|\; \mathcal{C}_N \right] \frac{\mathbf{P}(\mathcal{A}^B_0(\beta, \beta), g^*_{\lfloor \beta N \rfloor - 1} = 2 \mid \mathcal{C}_N)}{\mathbf{P}(g^*_{\lfloor \beta N \rfloor - 1} = 2 \mid \mathcal{C}_N)}$$

$$= \frac{(1 - \lambda)t + O(\sqrt{N \log N})}{t + \lambda(\beta N - t) + O(\sqrt{N \log N})} \cdot \frac{\lambda t + O(\sqrt{N \log N})}{\lambda \beta N + O(\sqrt{N \log N})} \cdot \frac{\mathcal{U}^B_{N, \lfloor \beta N \rfloor, 2}}{1 - \lambda + O(\frac{1}{N^2})}$$

$$= \frac{t^2}{((1 - \lambda)t + \lambda \beta N)\beta N} \mathcal{U}^B_{N, \lfloor \beta N \rfloor, 2} + O\left( \sqrt{\frac{\log N}{N}} \right).$$

$\square$

In the following lemma, we compute the exact limit of $\mathcal{U}^B_{N, \lfloor \beta N \rfloor, k}$ when the number of candidates goes to infinity.

**Lemma C.5.** *For all $B \geq 0$ and $k \in \{1, 2\}$,*

$$\mathcal{U}^B_{N, \lfloor \beta N \rfloor, k} = \lambda_k \beta^2 \sum_{b=0}^{B} \left( \frac{1}{\beta} - \sum_{\ell=0}^{b} \frac{\log(1/\beta)^\ell}{\ell!} \right) + O\left( \sqrt{\frac{\log N}{N}} \right).$$

*Proof.* By definition (2) of $\mathcal{U}^B_{N, t, k}$, we have

$$\mathcal{U}^B_{N, t, k} = \mathbf{P}(\mathcal{A}^B_{\lfloor \beta N \rfloor}(\alpha, \beta) \text{ succeeds}, g^*_{t-1} = k \mid \mathcal{C}_N),$$

and $\mathcal{A}^B_{\lfloor \beta N \rfloor}(\alpha, \beta)$ is simply the single-threshold algorithm with threshold $\beta N$ and budget $B$. Let $T = \lfloor \beta N \rfloor$. As in the proof of Lemma 3.1, we decompose the success probability of $\mathcal{A}^B_{\lfloor \beta N \rfloor}$ as follows

$$\mathcal{U}^B_{N, T, k} = \mathbf{P}(\mathcal{A}^B_T(\alpha, \beta) \text{ succeeds}, g^*_{T-1} = k \mid \mathcal{C}_N)$$

$$= \sum_{t=T}^{N} \sum_{\ell \in \{1, 2\}} \mathbf{P}(\mathcal{A}^B_T(\alpha, \beta) \text{ succeeds}, \rho_1 = t, g_t = \ell, g^*_{T-1} = k \mid \mathcal{C}_N)$$

$$= \sum_{t=T}^{N} \sum_{\ell \in \{1, 2\}} \left( \mathbf{P}(\mathcal{A}^B_T(\alpha, \beta) \text{ succeeds}, \rho_1 = t, R_t = 1, g_t = \ell, g^*_{T-1} = k \mid \mathcal{C}_N) \right.$$

$$\left. + \mathbf{P}(\mathcal{A}^B_T(\alpha, \beta) \text{ succeeds}, \rho_1 = t, R_t \neq 1, g_t = \ell, g^*_{T-1} = k \mid \mathcal{C}_N) \right).$$

The terms appearing in the sums above were computed in the proof of Lemma 3.1. It follows respectively from (4) and (7), with $K = 2$, that

$$\mathbf{P}(\mathcal{A}^B_T(\alpha, \beta) \text{ succeeds}, \rho_1 = t, R_t = 1, g_t = \ell, g^*_{T-1} = k \mid \mathcal{C}_N) = \frac{\lambda_\ell \lambda_k}{N} (T/t)^2 + O\left( \sqrt{\frac{\log N}{N^3}} \right),$$

$$\mathbf{P}(\mathcal{A}^B_T(\alpha, \beta) \text{ succeeds}, \rho_1 = t, R_t \neq 1, g_t = \ell, g^*_{T-1} = k \mid \mathcal{C}_N)$$

$$= \left( \frac{T^2}{t^3} + O\left( \sqrt{\frac{\log N}{N^3}} \right) \right) \mathbb{1}(B > 0, k \neq \ell) \mathcal{U}^{B-1}_{N, t+1, k},$$

where the $O$ terms are independent of $t$, they only depend on $\beta$. Therefore,

$$\mathcal{U}^B_{N, T, k} = \left( 1 + O\left( \sqrt{\frac{\log N}{N}} \right) \right) \sum_{t=T}^{N} \sum_{\ell \in \{1, 2\}} \left( \frac{\lambda_\ell \lambda_k}{N} (T/t)^2 + \frac{T^2}{t^3} \mathbb{1}(B > 0, k \neq \ell) \mathcal{U}^{B-1}_{N, t+1, k} \right)$$

$$= \left( 1 + O\left( \sqrt{\frac{\log N}{N}} \right) \right) \sum_{t=T}^{N} \left( \frac{\lambda_k}{N} (T/t)^2 + \frac{T^2}{t^3} \mathbb{1}(B > 0) \mathcal{U}^{B-1}_{N, t+1, k} \right).$$

The first sum can easily be computed

$$\sum_{t=T}^{N} \frac{\lambda_k}{N}(T/t)^2 = \frac{\lambda_k(T/N)^2}{N} \sum_{t=T}^{N} \frac{1}{(t/N)^2}$$

$$= \lambda_k(\beta^2 + O(\tfrac{1}{N})) \int_{\beta}^{1} \frac{du}{u^2} + O(\tfrac{1}{N})$$

$$= \lambda_k \beta(1 - \beta) + O(\tfrac{1}{N}) \,.$$

Therefore,

$$\mathcal{U}_{N,T,k}^{B} = \left(1 + O\left(\sqrt{\tfrac{\log N}{N}}\right)\right) \left(\lambda_k \beta(1 - \beta) + \mathbb{1}(B > 0) \sum_{t=T}^{N} \frac{T^2}{t^3} \mathcal{U}_{N,t+1,k}^{B-1} + O(\tfrac{1}{N})\right)$$

$$= \lambda_k \beta(1 - \beta) + \mathbb{1}(B > 0) \sum_{t=T}^{N} \frac{T^2}{t^3} \mathcal{U}_{N,t+1,k}^{B-1} + O\left(\sqrt{\tfrac{\log N}{N}}\right) \,.$$

Dividing by $\lambda_k$ yields

$$\left(\lambda_k^{-1} \mathcal{U}_{N,T,k}^{B}\right) = \beta(1 - \beta) + \mathbb{1}(B > 0) \sum_{t=T}^{N} \frac{T^2}{t^3} \left(\lambda_k^{-1} \mathcal{U}_{N,t+1,k}^{B-1}\right) + O\left(\sqrt{\tfrac{\log N}{N}}\right) \,.$$

thus the double-indexed sequence $(\lambda_k^{-1} \mathcal{U}_{N,t,k}^{b})_{b,t}$ satisfies the same recursion and initial condition as $(\mathbf{P}(\mathcal{A}_t^b(0,0) \text{ succeeds}))_{b,t}$ (see proof of Lemma 3.1) with $\beta$ instead of $w$ and $K = 2$. Therefore, we deduce immediately that:

$$\lambda_k^{-1} \mathcal{U}_{N,T,k}^{B} = \beta^2 \sum_{b=0}^{B} \left(\frac{1}{\beta} - \sum_{\ell=0}^{b} \frac{\log(1/\beta)^{\ell}}{\ell!}\right) + O\left(\sqrt{\tfrac{\log N}{N}}\right) \,.$$

$\square$

## C.3   Proof of Lemma 4.1

*Proof.* Using Lemmas C.1, C.2, C.3 and C.4 for $k = 2$, we obtain for all $t \in \{\lfloor \alpha N \rfloor, \ldots, \lfloor \beta N \rfloor - 1\}$

$$\mathcal{U}_{N,t,2}^{B} = \frac{\lambda}{N} \sum_{s=t}^{\lfloor \beta N \rfloor - 1} \left(\frac{(1 - \lambda)t^2}{((1 - \lambda)t + \lambda s)s} + O\left(\sqrt{\tfrac{\log N}{N}}\right)\right)$$

$$+ \frac{\mathbb{1}(B > 0)}{1 - \lambda} \sum_{s=t}^{\lfloor \beta N \rfloor - 1} \left(\frac{(1 - \lambda)\left((1 - \lambda)^2 t + \lambda(2 - \lambda)s\right)t^2}{((1 - \lambda)t + \lambda s)^2 s^2} + O\left(\sqrt{\tfrac{\log N}{N^3}}\right)\right) \mathcal{U}_{N,s+1,2}^{B-1}$$

$$+ \frac{t^2}{((1 - \lambda)t + \lambda \beta N)\beta N} \mathcal{U}_{N,\lfloor \beta N \rfloor,2}^{B} + O\left(\sqrt{\tfrac{\log N}{N}}\right) \,.$$

The $O$ terms inside the sums depend on the ratios $t/N$ and $s/N$, but using that $\alpha \leq t/N \leq s/N \leq 1$, it can be made only dependent on $\alpha$ and the other constants of the problem. Moreover, Thus we can

write

$$\mathcal{U}_{N,t,2}^B = \frac{\lambda(1-\lambda)t^2}{N} \sum_{s=t}^{\lfloor \beta N \rfloor - 1} \frac{1}{((1-\lambda)t + \lambda s)s}$$

$$+ \mathbb{1}(B > 0)\, t^2 \sum_{s=t}^{\lfloor \beta N \rfloor - 1} \frac{((1-\lambda)^2 t + \lambda(2-\lambda)s)}{((1-\lambda)t + \lambda s)^2 s^2} \mathcal{U}_{N,s+1,2}^{B-1}$$

$$+ \frac{t^2}{((1-\lambda)t + \lambda \beta N)\beta N} \mathcal{U}_{N,\lfloor \beta N \rfloor,2}^B + O\left(\sqrt{\tfrac{\log N}{N}}\right)$$

$$= \lambda(1-\lambda)\left(\tfrac{t}{N}\right)^2 \frac{1}{N} \sum_{s=t}^{\lfloor \beta N \rfloor - 1} \frac{1}{((1-\lambda)\frac{t}{N} + \lambda \frac{s}{N})\frac{s}{N}}$$

$$+ \mathbb{1}(B > 0)\left(\tfrac{t}{N}\right)^2 \frac{1}{N} \sum_{s=t}^{\lfloor \beta N \rfloor - 1} \frac{(1-\lambda)^2 \frac{t}{N} + \lambda(2-\lambda)\frac{s}{N}}{((1-\lambda)\frac{t}{N} + \lambda \frac{s}{N})^2 (\frac{s}{N})^2} \mathcal{U}_{N,s+1,2}^{B-1}$$

$$+ \frac{(t/N)^2}{((1-\lambda)\frac{t}{N} + \lambda \beta)\beta} \mathcal{U}_{N,\lfloor \beta N \rfloor,2}^B + O\left(\sqrt{\tfrac{\log N}{N}}\right).$$

Taking $t = \lfloor wN \rfloor = wN + O(1)$ and using Riemann sum convergence properties yields

$$\left(\tfrac{t}{N}\right)^2 \frac{1}{N} \sum_{s=t}^{\lfloor \beta N \rfloor - 1} \frac{1}{((1-\lambda)\frac{t}{N} + \lambda \frac{s}{N})\frac{s}{N}} = \frac{w^2}{N} \sum_{s=\lfloor wN \rfloor}^{\lfloor \beta N \rfloor - 1} \frac{1}{((1-\lambda)w + \lambda \frac{s}{N})\frac{s}{N}} + O\left(\tfrac{1}{N}\right)$$

$$= w^2 \int_w^\beta \frac{du}{((1-\lambda)w + \lambda u)u} + O\left(\tfrac{1}{N}\right)$$

$$= \frac{w}{1-\lambda}\left(\int_w^\beta \frac{du}{u} - \int_w^\beta \frac{du}{(1/\lambda - 1)w + u}\right) + O\left(\tfrac{1}{N}\right)$$

$$= \frac{w}{1-\lambda}\left(-\log(w/\beta) - \log(1 - \lambda + \lambda\beta/w)\right) + O\left(\tfrac{1}{N}\right)$$

$$= \frac{-w\log\left((1-\lambda)\frac{w}{\beta} + \lambda\right)}{1-\lambda} + O\left(\tfrac{1}{N}\right). \tag{11}$$

On the other hand,

$$\frac{(t/N)^2}{((1-\lambda)\frac{t}{N} + \lambda\beta)\beta} = \frac{w^2}{((1-\lambda)w + \lambda\beta)\beta} + O\left(\tfrac{1}{N}\right),$$

and Lemma C.5 gives for $k = 2$ that $\varphi_2^B(\alpha, \beta; \beta)$ exists, its expression is

$$\varphi_2^B(\alpha, \beta; \beta) = (1-\lambda)\beta^2 \sum_{b=0}^{B}\left(\frac{1}{\beta} - \sum_{\ell=0}^{b} \frac{\log(1/\beta)^\ell}{\ell!}\right), \tag{12}$$

and it satisfies $\mathcal{U}_{N,\lfloor \beta N \rfloor,2}^B = \varphi_2^B(\alpha, \beta; \beta) + O\left(\sqrt{\tfrac{\log N}{N}}\right)$. Consequently,

$$\mathcal{U}_{N,\lfloor wN \rfloor,2}^B = -\lambda w \log\left((1-\lambda)\tfrac{w}{\beta} + \lambda\right)$$

$$+ \mathbb{1}(B > 0)\frac{w^2}{N} \sum_{s=\lfloor wN \rfloor}^{\lfloor \beta N \rfloor - 1} \frac{(1-\lambda)^2 w + \lambda(2-\lambda)\frac{s}{N}}{((1-\lambda)w + \lambda \frac{s}{N})^2 (\frac{s}{N})^2} \mathcal{U}_{N,s+1,2}^{B-1}$$

$$+ \frac{w^2}{((1-\lambda)w + \lambda\beta)\beta} \varphi_2^B(\alpha, \beta; \beta) + O\left(\sqrt{\tfrac{\log N}{N}}\right). \tag{13}$$

Using this equality, we will prove by induction over $B \geq 0$ that, for all $w \in [\alpha, \beta]$, the limit $\varphi^B(\alpha, \beta; w) := \lim_{N\to\infty} \mathcal{U}_{N,\lfloor wN \rfloor,2}^B$ exists, is continuous, and satisfies

$$\mathcal{U}_{N,\lfloor wN \rfloor,2}^B = \varphi_2^B(\alpha, \beta; w) + O\left(\sqrt{\tfrac{\log N}{N}}\right).$$

**Initialization.** For $B = 0$ and $w \in [\alpha, \beta]$, (13) gives immediately

$$\mathcal{U}^B_{N, \lfloor wN \rfloor, 2} = -\lambda w \log \left((1 - \lambda)\tfrac{w}{\beta} + \lambda\right) + \frac{w^2}{((1 - \lambda)w + \lambda\beta)\beta} \varphi_2^B(\alpha, \beta; \beta) + O\left(\sqrt{\tfrac{\log N}{N}}\right). \quad (14)$$

**Induction.** Let $B \geq 1$, $w \in [\alpha, \beta]$, and assume that $\mathcal{U}^{B-1}_{N, \lfloor uN \rfloor, 2} = \varphi_2^B(\alpha, \beta; u) + O\left(\sqrt{\tfrac{\log N}{N}}\right)$ for all $u \in [\alpha, \beta]$, where the $O$ does not depend on $u$. Using this hypothesis for $u = \frac{s+1}{N}$, with $s \in \{t, \ldots, \lfloor \beta N \rfloor - 1\}$, along with te continuity of $\varphi_2^{B-1}(\alpha, \beta; \cdot)$ and Riemann sums convergence properties, yields

$$\frac{w^2}{N} \sum_{s = \lfloor wN \rfloor}^{\lfloor \beta N \rfloor - 1} \frac{(1 - \lambda)^2 w + \lambda(2 - \lambda)\frac{s}{N}}{((1 - \lambda)w + \lambda\frac{s}{N})^2(\frac{s}{N})^2} \mathcal{U}^{B-1}_{N, s+1, 2}$$

$$= \frac{w^2}{N} \sum_{s = \lfloor wN \rfloor}^{\lfloor \beta N \rfloor - 1} \frac{(1 - \lambda)^2 w + \lambda(2 - \lambda)\frac{s}{N}}{((1 - \lambda)w + \lambda\frac{s}{N})^2(\frac{s}{N})^2} \varphi_2^{B-1}(\alpha, \beta; \tfrac{s+1}{N}) + O\left(\sqrt{\tfrac{\log N}{N}}\right)$$

$$= w^2 \int_w^\beta \frac{(1 - \lambda)^2 w + \lambda(2 - \lambda)u}{((1 - \lambda)w + \lambda u)^2 u^2} \varphi_2^{B-1}(\alpha, \beta; u) du + O\left(\sqrt{\tfrac{\log N}{N}}\right).$$

Therefore, we deduce by (13) that

$$\mathcal{U}^B_{N, \lfloor wN \rfloor, 2} = -\lambda w \log \left((1 - \lambda)\tfrac{w}{\beta} + \lambda\right)$$

$$+ w^2 \int_w^\beta \frac{(1 - \lambda)^2 w + \lambda(2 - \lambda)u}{((1 - \lambda)w + \lambda u)^2 u^2} \varphi_2^{B-1}(\alpha, \beta; u) du$$

$$+ \frac{w^2}{((1 - \lambda)w + \lambda\beta)\beta} \varphi_2^B(\alpha, \beta; \beta) + O\left(\sqrt{\tfrac{\log N}{N}}\right).$$

This proves that $\mathcal{U}^B_{N, \lfloor wN \rfloor, 2} = \varphi_2^B(\alpha, \beta; w) + O\left(\sqrt{\tfrac{\log N}{N}}\right)$, where

$$\varphi_2^B(\alpha, \beta; w) = -\lambda w \log \left((1 - \lambda)\tfrac{w}{\beta} + \lambda\right)$$

$$+ w^2 \int_w^\beta \frac{(1 - \lambda)^2 w + \lambda(2 - \lambda)u}{((1 - \lambda)w + \lambda u)^2 u^2} \varphi_2^{B-1}(\alpha, \beta; u) du$$

$$+ \frac{w^2}{((1 - \lambda)w + \lambda\beta)\beta} \varphi_2^B(\alpha, \beta; \beta),$$

where the expression of $\varphi_2^B(\alpha, \beta; \beta)$ is given in (12). $\qquad \square$

## C.4 Proof of Lemma 4.2

*Proof.* Using Lemmas C.1, C.3, C.2 and C.4 for $k = 2$, we obtain for all $t \in \{\lfloor \alpha N \rfloor, \ldots, \lfloor \beta N \rfloor - 1\}$

$$
\mathcal{U}_{N,t,1}^{B} = \frac{\lambda}{N} \sum_{s=t}^{\lfloor \beta N \rfloor - 1} \left( \frac{\lambda t}{(1 - \lambda)t + \lambda s} + O\left(\sqrt{\tfrac{\log N}{N}}\right) \right)
$$

$$
+ \frac{\mathbb{1}(B > 0)}{1 - \lambda} \sum_{s=t}^{\lfloor \beta N \rfloor - 1} \left( \frac{\lambda^2 (1 - \lambda)(s - t)t}{((1 - \lambda)t + \lambda s)^2 s} + O\left(\sqrt{\tfrac{\log N}{N^3}}\right) \right) \mathcal{U}_{N,s+1,2}^{B-1}
$$

$$
+ \frac{t}{\beta N} \mathcal{U}_{N,\lfloor \beta N \rfloor, 1}^{B} + \frac{\lambda(\beta N - t)t}{((1 - \lambda)t + \lambda \beta N)\beta N} \mathcal{U}_{N,\lfloor \beta N \rfloor, 2}^{B} + O\left(\sqrt{\tfrac{\log N}{N}}\right)
$$

$$
= \frac{\lambda^2 (t/N)}{N} \sum_{s=t}^{\lfloor \beta N \rfloor - 1} \frac{1}{(1 - \lambda)\frac{t}{N} + \lambda \frac{s}{N}}
$$

$$
+ \mathbb{1}(B > 0) \frac{\lambda^2 (t/N)}{N} \sum_{s=t}^{\lfloor \beta N \rfloor - 1} \frac{\frac{s}{N} - \frac{t}{N}}{((1 - \lambda)\frac{t}{N} + \lambda \frac{s}{N})^2 \frac{s}{N}} \mathcal{U}_{N,s+1,2}^{B-1}
$$

$$
+ \frac{t/N}{\beta} \mathcal{U}_{N,\lfloor \beta N \rfloor, 1}^{B} + \frac{\lambda(\beta - \frac{t}{N})\frac{t}{N}}{((1 - \lambda)\frac{t}{N} + \lambda \beta)\beta} \mathcal{U}_{N,\lfloor \beta N \rfloor, 2}^{B} + O\left(\sqrt{\tfrac{\log N}{N}}\right) .
$$

Consider in the following $t = \lfloor wN \rfloor = wN + O(1)$. Using Riemann sums convergence properties, we have

$$
\frac{\lambda^2 (t/N)}{N} \sum_{s=t}^{\lfloor \beta N \rfloor - 1} \frac{1}{(1 - \lambda)\frac{t}{N} + \lambda \frac{s}{N}} = \lambda^2 w \int_w^\beta \frac{du}{(1 - \lambda)w + \lambda u} + O(\tfrac{1}{N})
$$

$$
= \lambda w \left[ \log((1 - \lambda)w + \lambda u) \right]_w^\beta + O(\tfrac{1}{N})
$$

$$
= \lambda w \log\left(1 - \lambda + \lambda \tfrac{\beta}{w}\right) + O(\tfrac{1}{N}) .
$$

Since $\mathcal{U}_{N,s,k}^{b} \leq 1$ for all $b, s, k$, and $t = wN + O(1)$, as in the proof of Lemma 4.1, we obtain

$$
\mathcal{U}_{N,\lfloor wN \rfloor, 1}^{B} = \lambda w \log\left(1 - \lambda + \lambda \tfrac{\beta}{w}\right) + O(\tfrac{1}{N})
$$

$$
+ \mathbb{1}(B > 0) \frac{\lambda^2 w}{N} \sum_{s=\lfloor wN \rfloor}^{\lfloor \beta N \rfloor - 1} \frac{\frac{s}{N} - w}{((1 - \lambda)w + \lambda \frac{s}{N})^2 \frac{s}{N}} \mathcal{U}_{N,s+1,2}^{B-1} + O(\tfrac{1}{N})
$$

$$
+ \frac{w}{\beta} \mathcal{U}_{N,\lfloor \beta N \rfloor, 1}^{B} + \frac{\lambda(\beta - w)w}{((1 - \lambda)w + \lambda \beta)\beta} \mathcal{U}_{N,\lfloor \beta N \rfloor, 2}^{B} + O\left(\sqrt{\tfrac{\log N}{N}}\right)
$$

$$
= \lambda w \log\left(1 - \lambda + \lambda \tfrac{\beta}{w}\right)
$$

$$
+ \mathbb{1}(B > 0) \frac{\lambda^2 w}{N} \sum_{s=\lfloor wN \rfloor}^{\lfloor \beta N \rfloor - 1} \frac{\frac{s}{N} - w}{((1 - \lambda)w + \lambda \frac{s}{N})^2 \frac{s}{N}} \mathcal{U}_{N,s+1,2}^{B-1}
$$

$$
+ \frac{w}{\beta} \varphi_1^{B}(\alpha, \beta; \beta) + \frac{\lambda(\beta - w)w}{((1 - \lambda)w + \lambda \beta)\beta} \varphi_2^{B}(\alpha, \beta; \beta) + O\left(\sqrt{\tfrac{\log N}{N}}\right) ,
$$

where we used Lemma C.5 in the last inequality, which guarantees that $\mathcal{U}_{N,\lfloor \beta N \rfloor, k}^{B} = \varphi_k^{B}(\alpha, \beta; \beta) + O\left(\sqrt{\tfrac{\log N}{N}}\right)$ for $k \in \{1, 2\}$, with

$$
\varphi_k^{B}(\alpha, \beta; \beta) = \lambda_k \beta^2 \sum_{b=0}^{B} \left( \frac{1}{\beta} - \sum_{\ell=0}^{b} \frac{\log(1/\beta)^\ell}{\ell!} \right) .
$$

Denoting by $\varphi^B(\alpha, \beta; \beta) = \varphi_1^B(\alpha, \beta; \beta) + \varphi_2^B(\alpha, \beta; \beta)$, i.e. $\varphi^B(\alpha, \beta; \beta) = \frac{1}{\lambda_k}\varphi_k^B(\alpha, \beta; \beta)$, we have

$$
\begin{aligned}
\frac{w}{\beta}\varphi_1^B(\alpha, \beta; \beta) + \frac{\lambda(\beta - w)w}{((1-\lambda)w + \lambda\beta)\beta}\varphi_2^B(\alpha, \beta; \beta) &= \left(\frac{\lambda w}{\beta} + \frac{\lambda(1-\lambda)(\beta - w)w}{((1-\lambda)w + \lambda\beta)\beta}\right)\varphi^B(\alpha, \beta; \beta) \\
&= \frac{\lambda w}{\beta}\left(1 + \frac{(1-\lambda)(\beta - w)}{(1-\lambda)w + \lambda\beta}\right)\varphi^B(\alpha, \beta; \beta) \\
&= \frac{\lambda w}{\beta}\left(\frac{\beta}{(1-\lambda)w + \lambda\beta}\right)\varphi^B(\alpha, \beta; \beta) \\
&= \frac{\lambda w}{(1-\lambda)w + \lambda\beta}\varphi^B(\alpha, \beta; \beta).
\end{aligned}
$$

Thus

$$
\begin{aligned}
\mathcal{U}_{N,\lfloor wN\rfloor,1}^B = {}& \lambda w \log\left(1 - \lambda + \lambda\frac{\beta}{w}\right) + \frac{\lambda w}{(1-\lambda)w + \lambda\beta}\varphi^B(\alpha, \beta; \beta) \\
& + \mathbb{1}(B > 0)\frac{\lambda^2 w}{N}\sum_{s=\lfloor wN\rfloor}^{\lfloor \beta N\rfloor - 1}\frac{\frac{s}{N} - w}{((1-\lambda)w + \lambda\frac{s}{N})^2\frac{s}{N}}\mathcal{U}_{N,s+1,2}^{B-1} + O\left(\sqrt{\frac{\log N}{N}}\right). \quad (15)
\end{aligned}
$$

Now, we will prove by induction over $B$ that $\mathcal{U}_{N,\lfloor wN\rfloor,1}^B = \varphi_1^B(\alpha, \beta; w) + O\left(\sqrt{\frac{\log N}{N}}\right)$ for all $w \in [\alpha, \beta]$, with $\varphi_1^B(\alpha, \beta; \cdot)$ a continuous function satisfying the recursion stated in the Lemma.

**Initialization** For $B = 0$, (15) yields immediately for all $w \in [\alpha, \beta]$

$$
\mathcal{U}_{N,\lfloor wN\rfloor,1}^B = \lambda w \log\left(1 - \lambda + \lambda\frac{\beta}{w}\right) + \frac{\lambda w}{(1-\lambda)w + \lambda\beta}\varphi^B(\alpha, \beta; \beta) + O\left(\sqrt{\frac{\log N}{N}}\right).
$$

**Induction** Let $B \geq 1$, and assume that $\mathcal{U}_{N,\lfloor uN\rfloor,1}^{B-1} = \varphi_1^{B-1}(\alpha, \beta; u) + O\left(\sqrt{\frac{\log N}{N}}\right)$ for all $u \in [\alpha, \beta]$, and that $\varphi_1^B(\alpha, \beta; \cdot)$ is continuous. Consequently, using Riemann sums convergence properties, it holds for all $w \in [\alpha, \beta]$ that

$$
\begin{aligned}
\frac{\lambda^2 w}{N}\sum_{s=\lfloor wN\rfloor}^{\lfloor \beta N\rfloor - 1}&\frac{\frac{s}{N} - w}{((1-\lambda)w + \lambda\frac{s}{N})^2\frac{s}{N}}\mathcal{U}_{N,s+1,2}^{B-1} \\
&= \frac{\lambda^2 w}{N}\sum_{s=\lfloor wN\rfloor}^{\lfloor \beta N\rfloor - 1}\frac{\frac{s}{N} - w}{((1-\lambda)w + \lambda\frac{s}{N})^2\frac{s}{N}}\left(\varphi_2^{B-1}(\alpha, \beta; \frac{s+1}{N}) + O\left(\sqrt{\frac{\log N}{N}}\right)\right) \\
&= \frac{\lambda^2 w}{N}\sum_{s=\lfloor wN\rfloor}^{\lfloor \beta N\rfloor - 1}\frac{\frac{s}{N} - w}{((1-\lambda)w + \lambda\frac{s}{N})^2\frac{s}{N}}\varphi_2^{B-1}(\alpha, \beta; \frac{s+1}{N}) + O\left(\sqrt{\frac{\log N}{N}}\right) \\
&= \lambda^2 w\int_w^\beta \frac{(u - w)\varphi_2^{B-1}(\alpha, \beta; u)}{((1-\lambda)w + \lambda u)^2 u}du + O\left(\sqrt{\frac{\log N}{N}}\right),
\end{aligned}
$$

thus, we have by substituting into (15)

$$
\begin{aligned}
\mathcal{U}_{N,\lfloor wN\rfloor,1}^B = {}& \lambda w \log\left(1 - \lambda + \lambda\frac{\beta}{w}\right) + \frac{\lambda w}{(1-\lambda)w + \lambda\beta}\varphi^B(\alpha, \beta; \beta) \\
& + \lambda^2 w\int_w^\beta \frac{(u - w)\varphi_2^{B-1}(\alpha, \beta; u)}{((1-\lambda)w + \lambda u)^2 u}du + O\left(\sqrt{\frac{\log N}{N}}\right) \\
= {}& \varphi_1^B(\alpha, \beta; w) + O\left(\sqrt{\frac{\log N}{N}}\right).
\end{aligned}
$$

$\square$

## C.5 Proof of Corollary 4.3.1

*Proof.* Assume that $\lambda \geq 1/2$. For any thresholds $0 < \alpha \leq \beta \leq 1$ Theorem 4.3 yields

$$\lim_{N \to \infty} \mathbf{P}(\mathcal{A}^B(\alpha, \beta) \text{ succeeds}) \geq \lambda\alpha \log\left(\tfrac{\beta}{\alpha}\right) + \alpha\beta S^B(\beta) .$$

We will now determine thresholds maximizing this lower bound. For a fixed $\beta$, we have

$$\frac{\partial}{\partial\alpha}\left(\lambda\alpha \log\left(\tfrac{\beta}{\alpha}\right) + \alpha\beta S^B(\beta)\right) \geq 0 \iff \lambda \log\left(\tfrac{\beta}{\alpha}\right) - \lambda + \beta S^B(\beta) \geq 0$$

$$\iff \lambda \log\left(\tfrac{\alpha}{\beta}\right) \leq \beta S^B(\beta) - \lambda$$

$$\iff \alpha \leq \frac{\beta}{e} \exp\left(\frac{\beta}{\lambda} S^B(\beta)\right) .$$

This proves that, for fixed $\beta$ the lower bound $\lambda\alpha \log\left(\tfrac{\beta}{\alpha}\right) + \alpha\beta S^B(\beta)$ is maximized on $[0, \beta]$ for $\alpha = \min(\beta, \frac{\beta}{e}\exp(\frac{\beta}{\lambda}S^B(\beta))) = h^B(\beta)$. With this choice of $\alpha$, the optimal choice of $\beta$ is the one maximizing the mapping $\beta \mapsto \lambda h^B(\beta) \log\left(\frac{\beta}{h^B(\beta)}\right) + h^B(\beta)\beta S^B(\beta)$.

In particular, for $B = 0$, we obtain $\tilde{\alpha}_0 = \lambda \exp(\frac{1}{\lambda} - 2)$ and $\tilde{\beta}_0 = \lambda$. They guarantee an asymptotic success probability of at least $\lambda^2 \exp(\frac{1}{\lambda} - 2)$. Given that the sequence $(S^B(w))_B$ is non-decreasing for all $w \in (0, 1]$, it holds for all $B \geq 0$ that

$$\lim_{N \to \infty} \mathbf{P}(\mathcal{A}^B(\tilde{\alpha}_B, \tilde{\beta}_B) \text{ succeeds}) \geq \lambda\tilde{\alpha}_B \log\left(\tfrac{\tilde{\beta}_B}{\tilde{\alpha}_B}\right) + \tilde{\alpha}_B\tilde{\beta}_B S^B(\tilde{\beta}_B)$$

$$= \max_{\alpha \leq \beta}\left\{\lambda\alpha \log\left(\tfrac{\beta}{\alpha}\right) + \alpha\beta S^B(\beta)\right\}$$

$$\geq \max_{\alpha \leq \beta}\left\{\lambda\alpha \log\left(\tfrac{\beta}{\alpha}\right) + \alpha\beta S^0(\beta)\right\}$$

$$= \lambda\tilde{\alpha}_0 \log\left(\tfrac{\tilde{\beta}_0}{\tilde{\alpha}_0}\right) + \tilde{\alpha}_0\tilde{\beta}_0 S^0(\tilde{\beta}_0)$$

$$= \lambda^2 \exp(\tfrac{1}{\lambda} - 2)$$

$$\geq \tfrac{1}{e} - (\tfrac{4}{e} - 1)\lambda(1 - \lambda) .$$

On the other hand, taking equal thresholds $\alpha = \beta = \frac{1}{e}$ then using Corollary 3.2.1 with $K = 2$ gives

$$\lim_{N \to \infty} \mathbf{P}(\mathcal{A}^B(\tilde{\alpha}_B, \tilde{\beta}_B) \text{ succeeds}) \geq \max_{\alpha \leq \beta}\left\{\lambda\alpha \log\left(\tfrac{\beta}{\alpha}\right) + \alpha\beta S^B(\beta)\right\}$$

$$\geq \frac{S^B(1/e)}{e^2}$$

$$\geq \frac{1}{e} - \frac{1}{e(B + 1)!} .$$

Thus, we deduce that

$$\lim_{N \to \infty} \mathbf{P}(\mathcal{A}^B(\tilde{\alpha}_B, \tilde{\beta}_B) \text{ succeeds}) \geq \frac{1}{e} - \min\left\{\frac{1}{e(B + 1)!}, (\tfrac{1}{e} - 1)\lambda(1 - \lambda)\right\} .$$

$\square$

## C.6 Proof of Theorem 4.3

*Proof.* By Lemmas A.3, 4.1 and 4.2, The success probability of Algorithm $\mathcal{A}(\alpha, \beta)^B$ can be written as

$$
\begin{aligned}
\mathbf{P}(\mathcal{A}^B(\alpha, \beta) \text{ succeeds}) &= \mathbf{P}(\mathcal{A}^B_{\lfloor \alpha N \rfloor}(\alpha, \beta) \text{ succeeds} \mid \mathcal{C}_N) + O(\tfrac{1}{N^2}) \\
&= \mathbf{P}(\mathcal{A}^B_{\lfloor \alpha N \rfloor}(\alpha, \beta) \text{ succeeds}, g^*_{\lfloor \alpha N \rfloor - 1} = 1 \mid \mathcal{C}_N) \\
&\quad + \mathbf{P}(\mathcal{A}^B_{\lfloor \alpha N \rfloor}(\alpha, \beta) \text{ succeeds}, g^*_{\lfloor \alpha N \rfloor - 1} = 2 \mid \mathcal{C}_N) + O(\tfrac{1}{N^2}) \\
&= \mathcal{U}^B_{N, \lfloor \alpha N \rfloor, 1} + \mathcal{U}^B_{N, \lfloor \alpha N \rfloor, 2} + O(\tfrac{1}{N^2}) \\
&= \varphi^B_1(\alpha, \beta; \alpha) + \varphi^B_2(\alpha, \beta; \alpha) + O\left(\sqrt{\tfrac{\log N}{N}}\right) \\
&= \lambda \alpha \log\left(1 - \lambda + \lambda \tfrac{\beta}{\alpha}\right) + \frac{\lambda \alpha \beta^2}{(1 - \lambda)\alpha + \lambda \beta} \sum_{b=0}^{B} \left(\frac{1}{\beta} - \sum_{\ell=0}^{b} \frac{\log(1/\beta)^\ell}{\ell!}\right) \\
&\quad + \mathbb{1}(B > 0) \lambda^2 \alpha \int_\alpha^\beta \frac{(u - \alpha)\varphi^{B-1}_2(\alpha, \beta; u)}{((1 - \lambda)\alpha + \lambda u)^2 u} du \\
&\quad - \lambda \alpha \log\left((1 - \lambda)\tfrac{\alpha}{\beta} + \lambda\right) + \frac{(1 - \lambda)\beta \alpha^2}{(1 - \lambda)\alpha + \lambda \beta} \sum_{b=0}^{B} \left(\frac{1}{\beta} - \sum_{\ell=0}^{b} \frac{\log(1/\beta)^\ell}{\ell!}\right) \\
&\quad + \mathbb{1}(B > 0) \alpha^2 \int_\alpha^\beta \frac{(1 - \lambda)^2 \alpha + \lambda(2 - \lambda)u}{((1 - \lambda)\alpha + \lambda u)^2 u^2} \varphi^{B-1}_2(\alpha, \beta; u) du + O\left(\sqrt{\tfrac{\log N}{N}}\right),
\end{aligned}
$$

then, regrouping the terms yields

$$
\begin{aligned}
\mathbf{P}(\mathcal{A}^B(\alpha, \beta) &\text{ succeeds}) \\
&= \lambda \alpha \log\left(1 - \lambda + \lambda \tfrac{\beta}{\alpha}\right) - \lambda \alpha \log\left((1 - \lambda)\tfrac{\alpha}{\beta} + \lambda\right) \\
&\quad + \frac{\alpha \beta}{(1 - \lambda)\alpha + \lambda \beta} (\lambda \beta + (1 - \lambda)\alpha) \sum_{b=0}^{B} \left(\frac{1}{\beta} - \sum_{\ell=0}^{b} \frac{\log(1/\beta)^\ell}{\ell!}\right) \\
&\quad + \mathbb{1}(B > 0) \alpha \int_\alpha^\beta \left(\lambda^2(1 - \tfrac{\alpha}{u}) + \tfrac{\alpha}{u}((1 - \lambda)^2 \tfrac{\alpha}{u} + \lambda(2 - \lambda))\right) \frac{\varphi^{B-1}_2(\alpha, \beta; u) du}{((1 - \lambda)\alpha + \lambda u)^2} + O\left(\sqrt{\tfrac{\log N}{N}}\right).
\end{aligned}
$$

Finally, observing that

$$
\begin{aligned}
(1 - \lambda)^2 \tfrac{\alpha}{u} + \lambda(2 - \lambda)) &= (1 - \lambda)^2 \tfrac{\alpha^2}{u^2} + 2\lambda(1 - \lambda)\tfrac{\alpha}{u} + \lambda^2 \\
&= \tfrac{1}{u^2}\left((1 - \lambda)\alpha + \lambda u\right)^2,
\end{aligned}
$$

we deduce the result

$$
\begin{aligned}
\mathbf{P}(\mathcal{A}^B(\alpha, \beta) &\text{ succeeds}) \\
&= \lambda \alpha \log\left(\tfrac{\beta}{\alpha}\right) + \alpha \beta \sum_{b=0}^{B} \left(\frac{1}{\beta} - \sum_{\ell=0}^{b} \frac{\log(1/\beta)^\ell}{\ell!}\right) + \mathbb{1}(B > 0) \alpha \int_\alpha^\beta \frac{\varphi^{B-1}_2(\alpha, \beta; u) du}{u^2} + O\left(\sqrt{\tfrac{\log N}{N}}\right),
\end{aligned}
$$

$\square$

## D   Optimal memory-less algorithm for two groups

In this section, we derive an optimal memoryless algorithm employing a dynamic programming approach. We analyze the state transitions depending on the algorithm's actions and the associated success probabilities for each state. Unlike previous sections, our study here is not asymptotic. Therefore, we do not rely on estimating the number of candidates in each group using concentration inequalities. Instead, we consider the exact number of candidates in each group as a parameter for decision-making at each step.

### D.1  Memoryless algorithms

One distinctive feature of Dynamic Threshold algorithms is their decision-making process, which solely depends on the observations at each step and the available budget, without recourse to past comparison history. We designate algorithms exhibiting this characteristic as memory-less algorithms.

**Definition D.1.** *An algorithm $\mathcal{A}$ for the $(K, B)$-secretary problem is memory-less if its actions at any step $t \in [N]$ depend only on the current observations $r_t, g_t, \mathbb{1}(R_t = 1)$, the available budget $B_t$, the cardinals $(|G_{t-1}^k|)_{k \in [K]}$.*

We assume that a memory-less algorithm is aware of the current step $t$ at any time, and knows the proportions of each group $(\lambda_k)_{k \in [K]}$. However, in our analysis, the knowledge of group proportions is dispensable since we investigate the asymptotic success probabilities of DT algorithms. Indeed, for setting thresholds that depend on group proportions, if the smallest threshold is at least $\epsilon > 0$, regardless of group proportions, it suffices to observe the first $\lfloor \epsilon N \rfloor$ candidates, then estimate $\bar{\lambda}_k = \lfloor \epsilon N \rfloor^{-1} \sum_{t=1}^{\lfloor \epsilon N \rfloor} \mathbb{1}(g_t = k)$ for all $k \in [K]$. The algorithm can choose the thresholds using $(\bar{\lambda}_k)_{k \in [K]}$ instead of $(\lambda_k)_{k \in [K]}$. As the number of candidates tends to infinity, $\bar{\lambda}_k$ becomes arbitrarily close to $\lambda_k$ with high probability, and so do the thresholds, assuming they are continuous functions of the group proportions. Though this introduces additional intricacies to the proofs, the fundamental proof arguments and the results remain the same. While Definition D.1 only includes deterministic algorithms, it can be easily extended to randomized algorithms, by considering the distributions of the actions instead of the actions themselves.

In the following lemma, we establish that the success probability of a memory-less algorithm, given the history up to step $t - 1$, is contingent upon only a few parameters, which are the available budget $B_t$, the group to which the best-observed candidate belongs $g^*t$, and the sizes of the groups $(|G_t^k|)k \in [K]$. Collectively, these parameters define the *state* of a memory-less algorithm at step $t$, which entirely determines the success probability of the algorithm starting from that state.

**Lemma D.1.** *For any memory-less algorithm $\mathcal{A}$ and $t \in [N]$, denoting by $\tau$ the stopping time of $\mathcal{A}$ and by $\mathcal{F}_{t-1}$ is the history of the algorithm up to step $t - 1$, i.e. the set of all the observations and actions taken by the algorithm until step $t - 1$, then*

$$\mathbf{P}(\mathcal{A} \text{ succeeds} \mid \tau \geq t, g_t^*, \mathcal{F}_{t-1}) = \mathbf{P}(\mathcal{A} \text{ succeeds} \mid \tau \geq t, g_t^*, B_t, (|G_t^k|)_{k \in [K]}) .$$

*Proof.* Let $\mathcal{A}$ be a memory-less algorithm, and let us denote by $\tau$ its stopping time. Conditionally to the history of the algorithm until step $t - 1$ and to the event $\{\tau \geq t\}$, the success probability of $\mathcal{A}$ depends on the future observations and the future actions of the algorithm.

Given that the algorithm is memory-less, at any step $s \geq t$, its actions $a_{s,1}, a_{s,2}$ depend on the observations $r_t, g_t, R_t$, the budget $B_t$ and $(|G_{s-1}^k|)_{k \in [K]}$.

Conditionally to the cardinals of the groups at step $t - 1$, the cardinals $(|G_{s-1}^k|)_{k \in [K]}$ are independent of the history $F_{t-1}$ because

$$|G_{s-1}^k| = |G_{t-1}^k| + \sum_{u=t}^{s-1} \mathbb{1}(g_u = k) \quad \forall k \in [K] ,$$

Moreover, since the candidates are observed in a uniformly random order, and the group memberships are also i.i.d random variables, then for all $s \geq 2$ distributions of $r_s, R_s$ depend only on the cardinals of each group at step $s - 1$, on $g_s$ and $g_{s-1}^*$. Also $g_s^*$ is a function of $g_{s-1}^*, g_s$ and $\mathbb{1}(R_s = 1)$:

$$g_s^* = \mathbb{1}(R_s \neq 1) g_{s-1}^* + \mathbb{1}(R_s \neq 1) g_s ,$$

and the budget $B_s$ satisfies

$$B_s = B_{s-1} - \mathbb{1}(a_{s-1,2} = \texttt{compare}) .$$

Therefore, Conditionally to the $B_t, (|G_{t-1}^k|)_{k \in [K]}, g_{t-1}^*$, the distributions of the observations and of the algorithm's actions at any step $s \geq t$ are independent of the history before step $t$. $\square$

At any given step $t$, a memory-less algorithm has access to the available budget $B_t$ and the number of previous candidates belonging to each group. In the case of two groups, this information reduces to

$(t, B_t, |G_{t-1}^1|)$, since $|G_{t-1}^2| = t - 1 - |G_{t-1}^1|$. The state of the algorithm, which fully determines its success probability, is given by the tuple $(t, B_t, |G_{t-1}^1|, g_{t-1}^*)$. However, $g_{t-1}^*$ is not known to the algorithm, hence it must make decisions relying on the limited information it has, to maximize the expected success probability, where the expectation is taken over $g_{t-1}^*$.

### D.2 State transitions

For any memory-less algorithm $\mathcal{A}$, we denote by $S_t(\mathcal{A})$ its state at step $t$, which is a tuple $(t, b, m, \ell)$. Here, $t - 1$ represents the count of previously rejected candidates, $b \geq 0$ denotes the available budget, $m < t$ indicates the number of prior candidates from group $G^1$, and $\ell \in \{1, 2\}$ is the group containing the best-seen candidate so far.

To examine the state transitions of the algorithm, it is imperative to first understand the distribution of the new observations at any given step $t$, depending on $S_t(\mathcal{A})$. While the group membership $g_t$ of candidate $x_t$ is independent of $S_t(\mathcal{A})$, both $r_t$ and $R_t$ are contingent on it.

**Lemma D.2.** *For any memory-less algorithm $\mathcal{A}$ and state $(t, m, b, \ell)$, denoting by $k = 3 - \ell$ the group index different from $\ell$, it holds that*

$$\mathbf{P}(r_t = 1 \mid S_t(\mathcal{A}) = (t, m, b, \ell), g_t = \ell) = \frac{1}{t}$$

$$\mathbf{P}(r_t = 1 \mid S_t(\mathcal{A}) = (t, m, b, \ell), g_t = k) = \frac{|G_{t-1}^k| + t}{t(|G_{t-1}^k| + 1)}$$

$$\mathbf{P}(R_t = 1 \mid S_t(\mathcal{A}) = (t, m, b, \ell), g_t = \ell, r_t = 1) = 1$$

$$\mathbf{P}(R_t = 1 \mid S_t(\mathcal{A}) = (t, m, b, \ell), g_t = k, r_t = 1) = \frac{|G_{t-1}^k| + 1}{|G_{t-1}^k| + t} \ ,$$

*where*

$$|G_{t-1}^k| = \begin{cases} m & \text{if } k = 1 \\ t - 1 - m & \text{if } k = 2 \end{cases} .$$

*Proof.* If $S_t(\mathcal{A}) = (t, m, b, \ell)$, then in particular $g_{t-1}^* = \ell$, i.e. $\max G_{t-1}^\ell > \max G_{t-1}^k$, thus

$$\begin{aligned}
\mathbf{P}(r_t = 1 \mid S_t(\mathcal{A}) = (t, m, b, \ell), g_t = \ell) &= \mathbf{P}(x_t > \max G_{t-1}^\ell \mid S_t(\mathcal{A}) = (t, m, b, \ell), g_t = \ell) \\
&= \mathbf{P}(x_t > \max x_{1:t-1} \mid S_t(\mathcal{A}) = (t, m, b, \ell), g_t = \ell) \\
&= \frac{1}{t} \ ,
\end{aligned}$$

because the rank of $x_t$ among previous candidates is independent of their relative ranks and groups, thus independent of the state of the algorithm. Moreover, if $r_t = 1$, $g_t = 1$ and $g_{t-1}^* = \ell$, then $x_t$ is the better than the maximum of $G_{t-1}^\ell$, which is the maximum of $x_{1:t-1}$, thus necessarily $R_t = 1$,

$$\mathbf{P}(R_t = 1 \mid S_t(\mathcal{A}) = (t, m, b, \ell), g_t = \ell, r_t = 1) = 1 \ .$$

On the other hand, if $g_t = k \neq \ell = g_{t-1}^*$, assume that $|G_{t-1}^\ell| > 0$. It holds that

$$\begin{aligned}
\mathbf{P}(r_t = 1 \mid S_t(\mathcal{A}) = (t, m, b, \ell), g_t = k) &= \mathbf{P}(r_t = 1 \mid g_{t-1}^* = \ell, g_t = k, |G_{t-1}^1| = m) \\
&= \frac{\mathbf{P}(r_t = 1, g_{t-1}^* = \ell \mid g_t = k, |G_{t-1}^1| = m)}{\mathbf{P}(g_{t-1}^* = \ell \mid |G_{t-1}^1| = m)} \ .
\end{aligned}$$

We have immediately that

$$\mathbf{P}(g_{t-1}^* = \ell \mid |G_{t-1}^1| = m) = \mathbf{P}(\max G_{t-1}^\ell > \max G_{t-1}^k \mid |G_{t-1}^1| = m) = \frac{|G_{t-1}^\ell|}{t - 1} \ ,$$

and the numerator can be computed as

$$\mathbf{P}(r_t = 1, g_{t-1}^* = \ell \mid g_t = k, |G_{t-1}^1| = m) \tag{16}$$

$$= \mathbf{P}(x_t > \max G_{t-1}^k, g_{t-1}^* = \ell \mid |G_{t-1}^1| = m)$$

$$= \mathbf{P}(x_t > \max G_{t-1}^k, \max G_{t-1}^\ell > \max G_{t-1}^k \mid |G_{t-1}^1| = m)$$

$$= \mathbf{P}(x_t > \max G_{t-1}^\ell > \max G_{t-1}^k \mid |G_{t-1}^1| = m)$$

$$+ \mathbf{P}(\max G_{t-1}^\ell > x_t > \max G_{t-1}^k \mid |G_{t-1}^1| = m)$$

$$= \frac{1}{t} \cdot \frac{|G_{t-1}^\ell|}{t-1} + \frac{|G_{t-1}^\ell|}{t} \cdot \frac{1}{|G_{t-1}^k| + 1}$$

$$= \frac{|G_{t-1}^\ell|}{t} \left( \frac{1}{t-1} + \frac{1}{|G_{t-1}^k| + 1} \right) , \tag{17}$$

which yields

$$\mathbf{P}(r_t = 1 \mid S_t(\mathcal{A}) = (t, m, b, \ell), g_t = k) = \frac{t-1}{|G_{t-1}^\ell|} \cdot \frac{|G_{t-1}^\ell|}{t} \left( \frac{1}{t-1} + \frac{1}{|G_{t-1}^k| + 1} \right)$$

$$= \frac{1}{t} \left( 1 + \frac{t-1}{|G_{t-1}^k| + 1} \right)$$

$$= \frac{|G_{t-1}^k| + t}{t(|G_{t-1}^k| + 1)} .$$

Finally,

$$\mathbf{P}(R_t = 1 \mid S_t(\mathcal{A}) = (t, m, b, \ell), g_t = k, r_t = 1) = \frac{\mathbf{P}(R_t = 1, r_t = 1, g_{t-1}^* = \ell \mid g_t = k, |G_{t-1}^1|)}{\mathbf{P}(r_t = 1, g_{t-1}^* = \ell \mid g_t = k, |G_{t-1}^1|)} .$$

We computed the denominator term in (17), and the numerator satisfies

$$\mathbf{P}(R_t = 1, r_t = 1, g_{t-1}^* = \ell \mid g_t = k, |G_{t-1}^1|) = \mathbf{P}(R_t = 1, g_{t-1}^* = \ell \mid g_t = k, |G_{t-1}^1|)$$

$$= \mathbf{P}(x_t > \max G^\ell G_{t-1}^\ell > \max G_{t-1}^k \mid |G_{t-1}^1|)$$

$$= \frac{1}{t} \cdot \frac{|G_{t-1}^\ell|}{t-1} ,$$

hence

$$\mathbf{P}(R_t = 1 \mid S_t(\mathcal{A}) = (t, m, b, \ell), g_t = k, r_t = 1) = \frac{\frac{|G_{t-1}^\ell|}{t(t-1)}}{\frac{|G_{t-1}^\ell|}{t} \left( \frac{1}{t-1} + \frac{1}{|G_{t-1}^k| + 1} \right)}$$

$$= \frac{1}{1 + \frac{t-1}{|G_{t-1}^k| + 1}}$$

$$= \frac{|G_{t-1}^k| + 1}{|G_{t-1}^k| + t} .$$

This concludes the proof when $|G_{t-1}^\ell| > 0$. If $|G_{t-1}^\ell| = 0$, then the same identities remain trivially true. □

Using the previous Lemma, we can fully characterize the possible state transitions of a memory-less algorithm. First, the values of the parameters $|G_t^1|$ and $B_{t+1}$ are trivially determined based on the state $S_t$ at the beginning of step $t$, the observations $r_t$ and $g_t$, and the actions of the algorithm:

$$|G_t^1| = |G_{t-1}^1| + \mathbb{1}(g_t = 1) , \quad B_{t+1} = B_t - \mathbb{1}(a_{t,1} = \texttt{compare}) ,$$

where $a_{t,1}$ is the action taken by the algorithm, which only depends on the state $S_t$ since the algorithm is memory-less.

Regarding $g_t^*$, if $g_t = g_{t-1}^*$, then $g_t^* = g_{t-1}^*$ remains unchanged with probability 1. However, if $g_t \neq g_{t-1}^*$ and $r_t = 1$, and if the algorithm skips the candidate without making a comparison, then $g_t^*$ is not deterministic based on the history alone. The probability that $g_t^* = g_t$ in this case is precisely the probability that $R_t = 1$, computed in Lemma D.2

$$\mathbf{P}(g_t^* = g_t \mid S_t(\mathcal{A}) = (t, m, b, \ell), g_t = k, r_t = 1) = \frac{|G_{t-1}^k| + 1}{|G_{t-1}^k| + t} .$$

### D.3 Expected action rewards

In the following, we denote by $\mathcal{A}_*$ the optimal memory-less algorithm for two groups, and for all $B \geq 0, t \in [N], m < t$ and $\ell \in \{1, 2\}$, we denote by

$$\mathcal{V}_{t,m,\ell}^B = \mathbf{P}(\mathcal{A}_* \text{ succeeds} \mid \tau \geq t, S_t(\mathcal{A}_*) = (t, B, m, \ell)) ,$$

which is its success probability starting from state $(t, B, m, \ell)$.

We analyze the expected rewards and state transitions of algorithm $\mathcal{A}_*$ given its limited information access. When the algorithm receives a new observation $(r_t, g_t)$:

- If $r_t \neq 1$, the optimal action is to skip the candidate (skip).
- If $r_t = 1$ and $B_t = 0$, the algorithm either stops or skips the candidate. However, if there is a positive budget $B_t$, stopping is suboptimal: it is always better to make a comparison first.
- If the algorithm chooses to make a comparison and observes $R_t$:
  - If $R_t \neq 1$, the optimal action is to skip the candidate.
  - If $R_t = 1$, the algorithm must decide whether to skip or stop. However, skipping after observing $R_t = 1$ is suboptimal compared to skipping immediately after observing $r_t = 1$, as the latter conserves the budget.

In summary, any rational algorithm follows these decision rules:

- If $(r_t \neq 1)$ or $(r_t = 1$ and $R_t \neq 1)$, then skip the candidate.
- If $(r_t = 1$ and $R_t = 1)$, select the candidate.

Therefore, the main non-trivial decision to make is whether to reject or accept a candidate after observing $r_t = 1$. Consider an algorithm $\mathcal{A}$ following these rules. At time $t$ with budget $B_t = b$ and $|G_{t-1}^1| = m$, if $g_t = k$ and $r_t = 1$, choosing an action $a \in \{\text{skip}, \text{stop}, \text{compare}\}$ based on these rules leads to a new state $S_{t+1}(\mathcal{A}) = F(t, b, m, k, a)$, which is a random variable depending on $g_{t-1}^*$ and $R_t$. If $a \in \{\text{stop}, \text{compare}\}$ and $R_t = 1$, then $S_{t+1}(\mathcal{A})$ is a final state: success or failure.

With this notation, we define $\mathcal{R}_{t,m}^B(a)$ as the reward that $\mathcal{A}_*$ expects to gain by playing action $a$ after observing $r_t = 1$ and $g_t = k$ in a state $S_t(\mathcal{A}_*) = (t, b, m, \cdot)$, where it ignores $g_{t-1}^*$

$$\mathcal{R}_{t,m,k}^B(a) = \mathbf{E}[\mathbf{P}(\mathcal{A}_* \text{ succeeds} \mid S_{t+1}(\mathcal{A}_*) = F(t, B, m, k, a)) \mid S_t(\mathcal{A}_*) = (t, B, m, \cdot), r_t = 1, g_t = k] .$$

where the expectation is taken over $g_{t-1}^*$ and $R_t$. The optimal memory-less action at any state $(t, B, m, \ell)$, knowing that $r_t = 1, g_t = k$, is the one maximizing $\mathcal{R}_{t,m,k}^B(a)$.

**Lemma D.3.** *Consider a state $S_t = (t, B, m, \cdot)$, and let $\{k, \ell\} = \{1, 2\}$, $M_k = m + \mathbb{1}(k = 1)$, then*

$$\mathcal{R}_{t,m,k}^B(\text{stop}) = \frac{|G_t^k|}{N} ,$$

$$\mathcal{R}_{t,m,k}^B(\text{skip}) = \frac{|G_t^1|}{t} \mathcal{V}_{t+1,M_k,1}^B + \frac{|G_t^2|}{t} \mathcal{V}_{t+1,M,2}^B ,$$

$$\mathcal{R}_{t,m,k}^B(\text{compare}) = \frac{|G_t^k|}{N} + \frac{|G_t^\ell|}{t} \left( \frac{|G_{t-1}^k| + 1}{|G_{t-1}^k| + t} \cdot \frac{t}{N} + \frac{t-1}{|G_{t-1}^k| + t} \mathcal{V}_{t+1,M_k,\ell}^{B-1} \right) ,$$

Observe that, conditionally to $g_t$ and $|G_{t-1}^1|$, the cardinals of $G_{t-1}^1, G_{t-1}^2, G_t^1, G_t^2$ are all known:

$$|G_t^1| = |G_{t-1}^1| + \mathbb{1}(g_t = 1) , \quad |G_t^2| = t - |G_t^1|, \quad |G_{t-1}^2| = t - 1 - |G_{t-1}^1| .$$

## D.4 Optimal actions and success probability

Using Lemma D.3 and considering the potential state transitions based on the actions, we establish a recursion satisfied by $(\mathcal{V}^B_{t,m,\ell})_{t,B,m,\ell}$. We present the result without distinction between the cases $\ell = 1$ and $\ell = 2$. For simplicity, let $\lambda_k = \mathbf{P}(g_t = k)$ for $k = 1, 2$, and define $M_k = m + \mathbb{1}(k = 1)$ for all $m \geq 0$. Additionally, for all $(B, t, m, k)$, define

$$\delta_k^B = \mathbb{1}\big(\mathcal{R}^B_{t,m,k}(\texttt{accept}) \geq \mathcal{R}^B_{t,m,k}(\texttt{skip})\big) \;,$$

where the action $\texttt{accept}$ corresponds to $\texttt{compare}$ for $B > 0$ and $\texttt{stop}$ for $B = 0$.

**Theorem D.4.** *For all $t \in [N]$, $m < t$ and $\{k, \ell\} = \{1, 2\}$, the success probability of $\mathcal{A}_*$ with zero budget satisfies the recursion*

$$\mathcal{V}^0_{t,m,\ell} = \lambda_\ell \left( \frac{\delta_\ell^0}{N} + \left(1 - \frac{\delta_\ell^0}{t}\right) \mathcal{V}^0_{t+1,M_\ell,\ell} \right)$$

$$+ \lambda_k \left( \frac{\delta_k^0}{N} + \frac{1 - \delta_k^0}{t} \mathcal{V}^0_{t+1,M_k,k} + \left(1 - \frac{1}{t}\right)\left(2 - \delta_k^0 - \frac{1}{|G^k_{t-1}|+1}\right) \mathcal{V}^0_{t+1,M_k,\ell} \right) \;,$$

*and for $B \geq 1$ it satisfies*

$$\mathcal{V}^B_{t,m,\ell} = \lambda_\ell \left( \frac{\delta_\ell^B}{N} + \left(1 - \frac{\delta_\ell^B}{t}\right)\mathcal{V}^B_{t+1,M_\ell,\ell} \right)$$

$$+ \lambda_k \left( \frac{\delta_k^B}{N} + \frac{\delta_k^B}{|G^k_{t-1}|+1}\left(1 - \frac{1}{t}\right)\mathcal{V}^{B-1}_{t+1,M_k,\ell} + \frac{1-\delta_k^B}{t}\mathcal{V}^B_{t+1,M_k,k} + \left(1 - \frac{1}{t}\right)\left(1 - \frac{\delta_k^B}{|G^k_{t-1}|+1}\right)\mathcal{V}^B_{t+1,M_k,\ell} \right) \;,$$

*where $\mathcal{V}^B_{N+1,m,k} = 0$ for all $B \geq 0$ $m \leq N$ and $k \in \{1, 2\}$.*

*Proof.* Using the results from Section D.2 and D.3, the actions of $\mathcal{A}_*$ and the resulting state transitions are as follows. If the state of $\mathcal{A}_*$ at step $t$ is $S_t(\mathcal{A}_*) = (t, B, m, \ell)$ for some $B \geq 1$, $m < t$ and $\ell \in \{1, 2\}$: If $g_t = \ell$, denoting by $M_\ell = m + \mathbb{1}(\ell = 1)$, we have

- with probability $1 - 1/t$: $r_t = 0$, and the algorithm rejects the candidate, transitioning to the state $(t + 1, B, M_\ell, \ell)$.

- with probability $1/t$: $r_t = 1$, and necessarily $R_t = 1$, because $g_t = g^*_{t-1} = \ell$.

  - If $\mathcal{R}^B_{t,m,k}(\texttt{compare}) > \mathcal{R}^B_{t,m,k}(\texttt{skip})$, then the algorithm uses a comparison and observes $R_t = 1$, hence accepts the candidate. The success probability in that case is $t/N$.
  - Otherwise, the candidate is rejected and the algorithm goes to state $(t + 1, B, M_\ell, \ell)$

On the other hand, if $g_t = k \neq g^*_{t-1}$, then denoting by $M_k = m + \mathbb{1}(k = 1)$, we have

- with probability $\frac{|G^k_{t-1}|(t+1)}{t(|G^k_{t-1}|+1)}$: $r_t = 0$, and the algorithm rejects the candidate, transitioning to the state $(t + 1, B, M_k, \ell)$.

- with probability $\frac{|G^k_{t-1}|+t}{t(|G^k_{t-1}|+1)}$: $r_t = 1$

  - If $\mathcal{R}^B_{t,m,k}(\texttt{compare}) > \mathcal{R}^B_{t,m,k}(\texttt{skip})$, then the algorithm uses a comparison
    * with probability $\frac{|G^k_{t-1}|+1}{|G^k_{t-1}|+t}$: $R_t = 1$ and the algorithm stops, its success probability is $t/N$
    * with probability $\frac{t-1}{|G^k_{t-1}|+t}$: $R_t = 0$, the candidate is rejected, and the algorithm goes to state $(t + 1, B - 1, M_k, \ell)$
  - Otherwise, the candidate is rejected and
    * with probability $\frac{|G^k_{t-1}|+1}{|G^k_{t-1}|+t}$: the algorithm goes to state $(t + 1, B, M_k, k)$
    * with probability $\frac{t-1}{|G^k_{t-1}|+t}$: the algorithm goes to state $(t + 1, B, M_k, \ell)$

In the case of a zero budget, the algorithm compares $\mathcal{R}^B_{t,m,k}(\texttt{skip})$ to $\mathcal{R}^B_{t,m,k}(\texttt{compare})$ instead of $\mathcal{R}^B_{t,m,k}(\texttt{compare})$. If the algorithm decides to reject the candidate then the same state transition occurs. However, if the candidate is selected and if $g_t = g^*_{t-1} = \ell$ then the success probability is $t/N$. If On the other hand, if it is selected and $g_t = k \neq g^*_{t-1} = \ell$ then the probability that the current candidate is the best overall is $\frac{|G^k_{t-1}|+1}{|G^k_{t-1}|+t} \times \frac{t}{N}$.

All in all, for $B = 0$, then

$$(\mathcal{V}^0_{t,m,\ell} \mid g_t = \ell) = \frac{1}{t}\left(\delta^0_\ell \frac{t}{N} + (1-\delta^0_\ell)\mathcal{V}^0_{t+1,M_\ell,\ell}\right) + \left(1 - \frac{1}{t}\right)\mathcal{V}^0_{t+1,M_\ell,\ell}$$

$$= \left(1 - \frac{\delta^0_\ell}{t}\right)\mathcal{V}^0_{t+1,M_\ell,\ell} + \frac{\delta^0_\ell}{N}$$

$$(\mathcal{V}^0_{t,m,\ell} \mid g_t = k) = \frac{|G^k_{t-1}|+t}{t(|G^k_{t-1}|+1)}\left(\delta^0_k \frac{t(|G^k_{t-1}|+1)}{N(|G^k_{t-1}|+t)} + (1-\delta^0_k)\left(\frac{|G^k_{t-1}|+1}{|G^k_{t-1}|+t}\mathcal{V}^0_{t+1,M_\ell,l} + \frac{t-1}{|G^k_{t-1}|+t}\mathcal{V}^0_{t+1,M_\ell,\ell}\right)\right)$$

$$+ \frac{|G^k_{t-1}|+t}{t(|G^k_{t-1}|+1)}\mathcal{V}^0_{t+1,M_k,\ell}$$

$$= \frac{\delta^0_k}{N} + \frac{1-\delta^0_k}{t}\mathcal{V}^0_{t+1,M_k,k} + \left(1-\frac{1}{t}\right)\left(2 - \delta^0_k - \frac{1}{|G^k_{t-1}|+1}\right)\mathcal{V}^0_{t+1,M_k,\ell} \,,$$

and we deduce that

$$\mathcal{V}^0_{t,m,\ell} = \lambda_\ell\left(\left(1 - \frac{\delta^0_\ell}{t}\right)\mathcal{V}^0_{t+1,M_\ell,\ell} + \frac{\delta^0_\ell}{N}\right)$$

$$+ \lambda_k\left(\frac{\delta^0_k}{N} + \frac{1-\delta^0_k}{t}\mathcal{V}^0_{t+1,M_k,k} + \left(1-\frac{1}{t}\right)\left(2 - \delta^0_k - \frac{1}{|G^k_{t-1}|+1}\right)\mathcal{V}^0_{t+1,M_k,\ell}\right)\,.$$

For $B \geq 1$, we obtain

$$(\mathcal{V}^B_{t,m,\ell} \mid g_t = \ell) = \frac{1}{t}\left(\delta^B_\ell \frac{t}{N} + (1-\delta^B_\ell)\mathcal{V}^B_{t+1,M_\ell,\ell}\right) + \left(1 - \frac{1}{t}\right)\mathcal{V}^B_{t+1,M_\ell,\ell}$$

$$= \left(1 - \frac{\delta^B_\ell}{t}\right)\mathcal{V}^B_{t+1,M_\ell,\ell} + \frac{\delta^B_\ell}{N}$$

$$(\mathcal{V}^B_{t,m,\ell} \mid g_t = k) = \frac{|G^k_{t-1}|+t}{t(|G^k_{t-1}|+1)}\left[\delta^B_k\left(\frac{|G^k_{t-1}|+1}{|G^k_{t-1}|+t}\mathcal{V}^B_{t+1,M_k,k} + \frac{t-1}{|G^k_{t-1}|+t}\mathcal{V}^B_{t+1,M_k,\ell}\right)\right.$$

$$\left. + (1-\delta^B_k)\left(\frac{|G^k_{t-1}|+1}{|G^k_{t-1}|+t}\mathcal{V}^0_{t+1,M_k,l} + \frac{t-1}{|G^k_{t-1}|+t}\mathcal{V}^0_{t+1,M_k,\ell}\right)\right] + \frac{|G^k_{t-1}|+t}{t(|G^k_{t-1}|+1)}\mathcal{V}^B_{t+1,M_k,\ell}$$

$$= \frac{\delta^B_k}{N} + \frac{\delta^B_k}{|G^k_{t-1}|+1}\left(1 - \frac{1}{t}\right)\mathcal{V}^{B-1}_{t+1,M_k,\ell} + \frac{1-\delta^B_k}{t}\mathcal{V}^B_{t+1,M_k,k} + \left(1 - \frac{1}{t}\right)\left(1 - \frac{\delta^B_k}{|G^k_{t-1}|+1}\right)\mathcal{V}^B_{t+1,M_k,\ell} \,,$$

hence

$$\mathcal{V}^B_{t,m,\ell} = \lambda_\ell\left(\frac{\delta^B_\ell}{N} + \left(1 - \frac{\delta^B_\ell}{t}\right)\mathcal{V}^B_{t+1,M_\ell,\ell}\right)$$

$$+ \lambda_k\left(\frac{\delta^B_k}{N} + \frac{\delta^B_k}{|G^k_{t-1}|+1}\left(1 - \frac{1}{t}\right)\mathcal{V}^{B-1}_{t+1,M_k,\ell} + \frac{1-\delta^B_k}{t}\mathcal{V}^B_{t+1,M_k,k} + \left(1 - \frac{1}{t}\right)\left(1 - \frac{\delta^B_k}{|G^k_{t-1}|+1}\right)\mathcal{V}^B_{t+1,M_k,\ell}\right)\,,$$

which concludes the proof. □

Implementing the optimal memory-less algorithm $\mathcal{A}_*$ with budget $B$ requires knowing the $(\mathcal{R}^b_{t,m,k}(a))_{t,b,m,k}$ for $a \in \{\texttt{skip}, \texttt{stop}, \texttt{compare}\}$, which depend themselves on the table $(\mathcal{V}^b_{t,m,k})_{t,b,m,k}$. Using Lemma D.3 and Theorem D.4, these tables can be computed in a $O(BN^2)$ time as described in Algorithm 2.

After computing these tables, the optimal memory-less algorithm $\mathcal{A}_*$ can be implemented by following the rational decision rules outlined in Section D.3, and when encountering $r_t = 1$ and needing to choose between accepting or rejecting the candidate, $\mathcal{A}_*$ selects the action that maximizes its expected reward given the information it has about the current state. A detailed description is provided in Algorithm 3.

**Algorithm 2:** $(\mathcal{V}^b_{t,m,k})_{t,b,m,k}$ and $(\mathcal{R}^b_{t,m,k}(a))_{t,b,m,k}$ for $a \in \{\texttt{skip}, \texttt{stop}, \texttt{compare}\}$

---

**Input:** Number of candidates $N$, available budget $B$, probability distribution of $g_t$: $\lambda_1, \lambda_2$
**Initialization:** $\mathcal{V}^b_{N+1,m,k} \leftarrow 0$ for all $b \leq B, m \leq N, k \in \{1,2\}$

**1** **for** $b = 1, \ldots, B$ **do**
**2**     **for** $t = N, N-1, \ldots, 1$ **do**
**3**        **for** $m = 0, \ldots, t$ **do**
**4**           Compute $\mathcal{R}^b_{t,m,k}(a)$ for $k \in \{1,2\}$ and $a \in \{\texttt{skip}, \texttt{stop}, \texttt{compare}\}$ using Lemma D.3
**5**           Compute $\mathcal{V}^b_{t,m,k}$ for $k \in \{1,2\}$ using Theorem D.4
**6** Return: $(\mathcal{V}^b_{t,m,k})_{t,b,m,k}, (\mathcal{R}^b_{t,m,k}(a))_{t,b,m,k}$

---

**Algorithm 3:** Optimal memory-less algorithm $\mathcal{A}_*$

---

**Input:** Number of candidates $N$, available budget $B$, probability distribution of $g_t$: $\lambda_1, \lambda_2$
**Initialization:** $b \leftarrow B, m \leftarrow 0$
**1** Compute $(\mathcal{V}^b_{t,m,k})_{t,b,m,k}$ and $(\mathcal{R}^b_{t,m,k}(a))_{t,b,m,k,a}$ using Algorithm 2
**2** **for** $t = 1, \ldots, N$ **do**
**3**     Receive new observation $(r_t, g_t)$
**4**     **if** $r_t = 1$ **then**
**5**        **if** $b = 0$ *and* $\mathcal{R}^0_{t,m,g_t}(\texttt{stop}) > \mathcal{R}^0_{t,m,g_t}(\texttt{skip})$ **then**
**6**           Return: $t$
**7**        **if** $b > 0$ *and* $\mathcal{R}^b_{t,m,g_t}(\texttt{compare}) > \mathcal{R}^b_{t,m,g_t}(\texttt{skip})$ **then**
**8**           $b \leftarrow b - 1$
**9**           **if** $R_t = 1$ **then**
**10**             Return: t
**11**     $m \leftarrow m + \mathbb{1}(g_t = 1)$

---

