# OpenReview forum: "Addressing Bias in Online Selection with Limited Budget of Comparisons"
_NeurIPS.cc/2024/Conference — NeurIPS 2024 poster_

### Official Review · Reviewer_io3s · 2024-07-05

**Soundness:** 3
**Presentation:** 4
**Contribution:** 3
**Rating:** 7
**Confidence:** 3

**Summary:**

The paper studies an extension of the online secretary problems in which candidates from multiple groups arrive. The goal is to find the best candidate, but while within-group comparisons are free, inter-group comparisons are not available a priori and we have a limit of B queries to a comparison oracle for such comparisons.
The authors study the single threshold algorithm in which we wait up until some time t and then either selects the first candidate if its the best among all groups so far (by doing a comparison) or if comparisons have been used up selects a candidate if it is the best candidate in its group seen so far.  The authors derive a lower bound on the probability of success that converges rapidly to 1/e as the budget B tends to infinity. This algorithm is independent of the group proportions. For the case of two groups, they analyse a double threshold algorithm and show a more fine-grained analysis that does depend on the group proportions. They further provide the optimal memoryless algorithm for the case of two groups, facilitating the computation of the optimal bounds. They show that the double threshold algorithm fares well in comparison and that the optimal memoryless algorithm behaves like a double threshold algorithm in the large candidate limit.

**Strengths:**

The idea of costly comparisons is interesting and the authors provide an interesting initial model to study this problem.

I think this is a thorough and very well written paper.  Although the results pertain to special case, they seem difficult to obtain.
The numerical experiments are helpful and give insights into the problem, that may be hard to prove theoretically, such as Figure 4.

**Weaknesses:**

The focus on the case of two groups is rather restrictive. Furthermore, it is unclear whether the techniques would generalise to further groups, or very much rely on this special case. In any case, the case for more groups seems non-trivial.

The discussion on the alternate model is missing from the main body and appendix, despite available space in the main body.

The figures would benefit from an explanation of what the dashed lines and what the continuous lines each represent.
Detailed/minor comments:

The sentence on line 36 is incomplete.

Line 93: cardinal -> cardinality

Line 148-151: Make the notation consistent with the pseudocode (whichever way around).

Line 110: full stop missing

Figure 2 and line? Why does the second row in Figure 2 not have (1-\lambad)t displayed instead?

256: cardinal-> cardinality
What is |G_t^{1/2}|

Line 259-260: The phrasing is unclear.

**Questions:**

2. 103-104: While I appreciate listing two possible models (indeed, I think he first one seems more natural), you claim here that throughout the paper you discuss how to extend the results to model 1 (which to me appears more natural). However, reading the main body I didn’t see this. Can you please explain what the results are and specify where in the paper you discuss the first model?

**Limitations:**

No concerns.

---

> ### Author Rebuttal · Authors · 2024-08-03
>
> We thank the reviewer for their feedback on our submission. We fully agree with the positive evaluation made, both in terms of strengths (solid theoretical study and introduction of a novel and realistic variant of the multi-color secretary problem) and in terms of future work (rigorous theoretical study of the multiple group case). We provide a detailed reply to the weaknesses and questions raised below.
> * **Focus on two groups.** The single threshold algorithm is studied for multiple groups, and the analysis shows how the dependence between $K$ and $B$ impacts the success probability. For algorithms with group-dependent thresholds, the analysis with two groups is already technically challenging and necessitates heavy computations. The analysis for an arbitrary number of groups $K$ should be technically possible. We believe that this study constitutes a logical future work for a journal version of this paper.
> * **Minor comments.** We thank the reviewers for pointing out the typos. We went carefully through the paper and corrected all of them.
>     * Regarding Figure 2, we preferred to display both acceptance regions depending on $\lambda t$ because the dynamic programming algorithm uses as a parameter the number of observed elements in $|G^1_t|$; $|G^2_t|$ is implicitly given by $t - |G^1_t|$. We believe that the current figure gives a good intuitive understanding of the inner logic of the algorithm.
>     * $|G_t^{1/2}|$ should be $|G_t^{1}|$ instead. We corrected it.
> * **Alternative model.** The single threshold algorithm with $K$ groups can be adapted for the first comparison model at a cost of $K-1$ additional costly comparisons. After the first $\alpha N$  candidates are rejected, $K-1$ comparisons are made between the maximum candidates from each group to identify the best candidate so far. The algorithm then keeps track of this best candidate. Whenever a new candidate becomes the best in their group, they can be compared to the current best candidate using a single comparison and the latter is updated accordingly. This approach enjoys the same guarantees as in Theorem 3.2, but with a budget of $K + B - 1$ instead of $B$.
>
>     In the case of two groups, both models are equivalent, as freely comparing a candidate with the best in their group and then making a costly comparison with the best candidate from the other group is sufficient to determine if they are the best so far. Thus, the guarantees on algorithms for the case of two groups remain the same.
>
>     This discussion was removed from the paper due to page limits and it was not included in the appendix by mistake.

---

> ### Comment · Reviewer_io3s · 2024-08-14
>
> Thank you for your response. Regarding the alternate model: You seem to have some space available still in your main body, so it would be worth including this as a brief discussion. Please also clarify in an update version of the paper what the lines in your figures mean, i.e. what do dashed and continuous lines each represent. I am updating my score.

---

> > ### Author Response · Authors · 2024-08-14
> >
> > We thank the reviewer again for their feedback and appreciate the positive reevaluation of the score.
> >
> > In Figure 3, the continuous and dotted lines represent respectively the success probabilities of the optimal memory-less algorithm $\mathcal{A}_*$ and the algorithm of Corollary 4.3.1. In Figure 4, they represent respectively the success probabilities of the DT algorithm with optimal thresholds and the optimal memory-less algorithm.
> >
> > We will include this clarification, along with a discussion on the alternate model, in the revised version of the paper.

---

### Official Review · Reviewer_5bhX · 2024-07-12

**Soundness:** 4
**Presentation:** 3
**Contribution:** 4
**Rating:** 6
**Confidence:** 4

**Summary:**

This paper studies a novel extension of multi-color secretary problem, where comparing candidates from different groups is possible at a cost. With a limited budget total, a Dynamic-Threshold algorithms family is introduced, and the success probability of a special case, i.e. single-threshold algorithm for K groups, is comprehensively analyzed. Moreover, in-depth theories have been studied in the scenario of two groups, including double threshold algorithms and the optimal memory-less algorithm.

**Strengths:**

1. This paper investigates a novel variant of the multi-color secretary problem that allows comparisons between candidates from different groups at a certain cost. This problem setting is appropriate for some real-world recruitment scenarios.

2. The algorithms proposed in this article are supported by solid theoretical results and are simple to implement yet effective.

3. This paper is well-organized and presented in a concise, clear and fluent manner.

**Weaknesses:**

1. The title of this paper is not quite appropriate, since bias defined in the field of statistics does not seem to exist in the multi-color secretary problem. I understand that there are different definitions of bias in real-life scenarios, but this could mislead readers in various areas.
2. There are some mistakes in the presentation of this article. For example, line 36 is not complete. Besides, the citation form in line 73 is not correct.

**Questions:**

I would like to know whether the theories in Section 4 can be generalized to the case of more than 2 groups. If so, is the generalization straightforward?

**Limitations:**

The authors adequately discussed the limitations of the paper.

---

> ### Author Rebuttal · Authors · 2024-08-03
>
> We thank the reviewer for their feedback on our submission. We fully agree with the positive evaluations made, both in terms of strengths (solid theoretical study and introduction of a novel and realistic variant of the multi-color secretary problem) and in terms of future work (rigorous theoretical study of the multiple group case). We provide a detailed reply to the weaknesses and questions raised below.
> * **Weakness 1.** The title of the paper is meant to reference the paper "Fairness and Bias in Online Selection" in ICML 2021. The bias here refers to the difficulty of accurately comparing candidates from different groups and our paper suggests that these comparisons can be made at a cost.
> * **Weakness 2.** We thank the reviewer for this observation. We went carefully through the paper and corrected all the typos and presentation mistakes.
> * **Question.** The analysis presented in Section 4 can be generalized to multiple groups, but this would significantly increase the overall complexity as the computations are already cumbersome and technically challenging for the case of two groups. Additionally, the results would become much more difficult to interpret and/or visualize (e.g., Figure 2 would be way heavier). For the aforementioned reasons we decided to postpone this study for a journal version of the paper, yet we agree that this constitutes a logical direction for future work.

---

> ### Comment · Reviewer_5bhX · 2024-08-13
>
> Thanks for the rebuttal from the authors. I have no further questions. After reading the paper again and all the reviews, I think my previous rating is adequate.

---

### Official Review · Reviewer_U5mj · 2024-07-17

**Soundness:** 3
**Presentation:** 3
**Contribution:** 3
**Rating:** 5
**Confidence:** 2

**Summary:**

This paper tackles an online hiring selection with budget problem. The authors propose a dynamic threshold method and provides theoretical analysis on the algorithm performance. The numerical experiments confirm the findings of the algorithm.

**Strengths:**

1. This paper is well motivated, and the proposed method is technical sound.
2. Related work is extensively discussed and surveyed.
3. Extensive and rigorous theoretical analysis is provided.

**Weaknesses:**

1. Although the problem is well motivated, it lacks of empirical analysis on real datasets and applications.
2. No baseline methods are compared in the experiments.

**Questions:**

See above weakness.

**Limitations:**

See above weakness.

---

> ### Author Rebuttal · Authors · 2024-08-03
>
> We thank the reviewer for their feedback on our submission. We generally agree with the positive evaluations made in terms of strengths (solid theoretical study and introduction of a novel and realistic variant of the multi-color secretary problem). We provide a detailed reply to the weaknesses and questions raised below.
> * **Real datasets.** In the secretary problem, the actual values of the items observed by the algorithm are irrelevant; only their relative order matters. Consequently, conducting experiments on real datasets is equivalent to using synthetic data. The key factors to consider are the budget, group proportions, and the number of candidates. We conducted experiments with various values of these parameters for two groups and additional experiments involving multiple groups are presented in Appendix A. We plan to include these experiments in the main body of the paper upon acceptance, as an additional page is allowed.
> * **Baseline methods.** Since the multi-color secretary problem has not been previously studied with budget constraints, the only available baselines are the algorithms proposed by Correa et al. (ICML 2021), which are optimal for the case of zero budget, and the classical $1/e$ strategy for an infinite budget. In our experiments, we compare our algorithms to the success probability of $1/e$ and the optimal algorithms for a zero-budget scenario.

---

### Decision · Program_Chairs · 2024-09-25

**Decision:**

Accept (poster)

**Comment:**

This paper was interesting. Two of the reviewers gave positive but short reviews and then did not respond to the authors' rebuttal. The responsive reviewer updated their score to a 7 after rebuttal. I stepped in and read the paper myself and asked clarifying questions to authors. My initial review (just seen by me) is below:

	I am reasonably impressed by this paper. I list strengths and weaknesses below, but I feel the theoretical strengths outweigh the empirical limitations of the work, even given the simple two-group framework they study. I'd push for weak acceptance.

	Strengths:
	(+) Clear motivation, simple theoretical framework, well-written
	(+) Nontrivial theoretical results even for relatively simple cases (i.e., two groups with potentially different thresholds or a single threshold for multiple groups)
	(+) I appreciated the authors' clarification that the two models are essentially the same K-1 comparisons to initially establish a "best over all groups" candidate that then gets updated as the process progresses
	(+) I agree with the authors' rebuttal that reviewers' concerns about real-world data are overblown

	Weaknesses:
	(-) I appreciated the authors' suggestion in their rebuttal that their machinery would extend to more than two groups; however, I am somewhat suspicious that it would be more complicated than they made it seem
	(-) The figures in the empirical section are poorly labeled and explained
	(-) Would it be possible to obtain any meaningful empirical results (even the performance of reasonable heuristics?) for more than two groups to give a taste of what extending results past two groups may look like? In a similar vein, is it possible to add any "natural" baselines in the case of two groups? Maybe something as simple as using the single-threshold algorithm would be a useful comparison point
	(-) In the vein of picking on experiments, would it be possible to examine what happens for higher values of B? B = 2 seems like a very small maximum budget
	(-) Minor typos throughout; largely did not influence my understanding of the paper

The authors presented a nice response to W3 (baseline) and W4 (B = 2 budget). They did acknowledge that the extension to more than two groups could be challenging, but they said that they were still confident that they could derive expressions in more complicated settings.